# HOComp: Interaction-Aware Human-Object Composition

**Dong Liang**
Tongji University / CityUHK / HKUST(GZ)
sse_liangdong@tongji.edu.cn

**Jinyuan Jia**[†]
Tongji University / HKUST(GZ)
jinyuanjia@hkust-gz.edu.cn

**Yuhao Liu**[†]
CityUHK
yuhaoliu7456@gmail.com

**Rynson W.H. Lau**[†]
CityUHK
Rynson.Lau@cityu.edu.hk

## Abstract

While existing image-guided composition methods may help insert a foreground object onto a user-specified region of a background image, achieving natural blending inside the region with the rest of the image unchanged, we observe that these existing methods often struggle in synthesizing seamless interaction-aware compositions when the task involves human-object interactions. In this paper, we first propose ***HOComp***, a novel approach for compositing a foreground object onto a human-centric background image, while ensuring harmonious interactions between the foreground object and the background person and their consistent appearances. Our approach includes two key designs: (1) ***MLLMs-driven Region-based Pose Guidance (MRPG)***, which utilizes MLLMs to identify the interaction region as well as the interaction type (*e.g.*, holding and lefting) to provide coarse-to-fine constraints to the generated pose for the interaction while incorporating human pose landmarks to track action variations and enforcing fine-grained pose constraints; and (2) ***Detail-Consistent Appearance Preservation (DCAP)***, which unifies a shape-aware attention modulation mechanism, a multi-view appearance loss, and a background consistency loss to ensure consistent shapes/textures of the foreground and faithful reproduction of the background human. We then propose the first dataset, named *Interaction-aware Human-Object Composition (IHOC)*, for the task. Experimental results on our dataset show that ***HOComp*** effectively generates harmonious human-object interactions with consistent appearances, and outperforms relevant methods qualitatively and quantitatively. Project page: `https://dliang293.github.io/HOComp-project/`.

## 1 Introduction

Considering a scenario in which a designer aims to create a perfume advertisement by compositing the image of a product onto an existing photograph with a human person, as shown in row 1 of Fig. 1, two critical objectives need to be satisfied in order to produce a visually convincing output. First, the interaction between the person and the perfume bottle should appear ***natural***, such that the bottle may seem to be appropriately related to (*e.g.*, held by) the person. Second, visual ***consistency*** must be maintained, preserving the original identities of both the person (including facial features and makeup) and the perfume bottle (*e.g.*, the logo, color, and shape).

---

[†]Joint corresponding authors.

39th Conference on Neural Information Processing Systems (NeurIPS 2025).

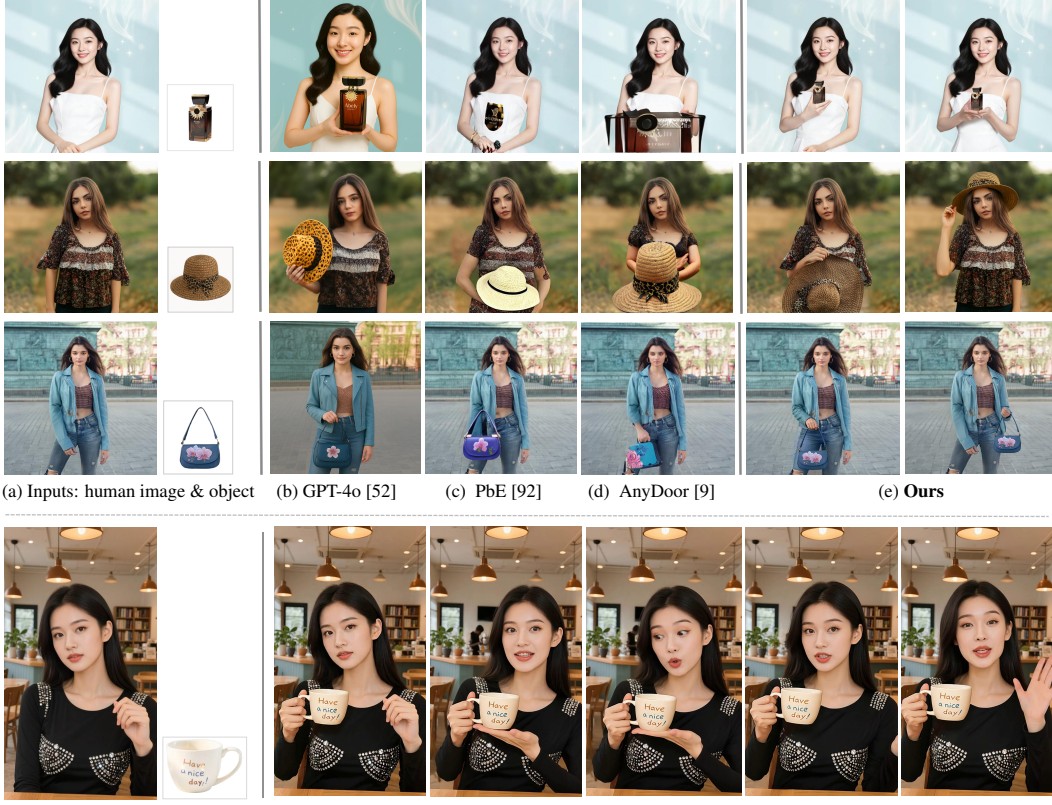

(a) Inputs: human image & object    (b) GPT-4o [52]    (c) PbE [92]    (d) AnyDoor [9]    (e) **Ours**

(a) Inputs: human image & object        (f) Generated video frames

Figure 1: When compositing a foreground object onto a human-centric background image, existing methods (b-d) typically rely on manually specifying the target region and text prompt, and often produce unrealistic interactions and inconsistent foreground/background appearances. In contrast, our proposed **HOComp**, automatically identifies the target region and generates a suitable text prompt to guide the interaction, resulting in realistic, harmonious and diverse interactions. Note that the text prompts used by the existing methods in the above three examples are: "A model is showing a perfume bottle", "A girl is holding a hat", and "A woman is lifting a handbag". By integrating with an Image-to-Video (I2V) model, our approach can support applications like human-product demonstration video generation (see results on the bottom region).

Some existing image-guided composition tasks [90, 37, 83] may be most relevant to the above task setting. They take a user-supplied foreground exemplar, typically accompanied by a textual prompt and a user-defined target region, and aim to synthesize a harmonious composition. Within this paradigm, they either incorporate identity-preservation modules [9, 68] to explicitly retain the original foreground details or focus on adjusting the colors, shadows, and perspective of the foreground to harmonize it with the background [49, 70, 92, 67], thereby producing photorealistic compositions. Despite the success, when the composition involves human and object interactions, as depicted in Fig. 1, existing methods [9, 67, 92] struggle to produce satisfactory results.

For our composition task, we observe that existing methods tend to fail in one or both of the following ways: (1) they may produce inappropriate gestures for the background persons (*e.g.*, most results in Fig. 1(c,d)); and (2) they may change the contents/identities of the foreground objects (*e.g.*, rows 2 and 3 of Fig. 1(b-d)) and/or the background persons (*e.g.*, the face in row 1 of Fig. 1(b), and the clothes in row 2 of Fig. 1(b,c) and row 3 of Fig. 1(b,d). To address these problems, we propose **HOComp**, an interaction-aware human-object composition framework, to create seamless composited images with harmonious human-object interactions and consistent appearances.

Our **HOComp** includes two key designs. The first design is the *MLLMs-driven region-based pose guidance (MRPG)*, which aims to constrain the human-object interaction. By utilizing the capabilities

of MLLMs, our method automatically determines suitable interaction types [2] (*e.g., holding, eating*) and interaction region. Here, we adopt a *coarse-to-fine constraint strategy*. We first use the interaction region generated by MLLMs as a coarse-level constraint to restrict the region of the background image for the interaction. We then incorporate human pose landmarks as a supervision to capture the variation of the human pose in the interaction, providing a fine-grained constraint on the pose within the interaction region. The second design is the *detail-consistent appearance preservation (DCAP)*, which aims to ensure foreground/background appearance consistency. For the foreground object, we propose a shape-aware attention modulation mechanism to explicitly manipulate attention maps for maintaining a consistent object shape, and a multi-view appearance loss to further preserve the object textures at the semantic level. For the background image, we propose a background consistency loss to retain the details of the background person outside the interaction region.

To train the model, we introduce a new dataset called *Interaction-aware Human-Object Composition (IHOC) dataset*, which includes images of humans before and after interacting with the foreground object, the interaction region, and the corresponding interaction type. We conduct extensive experiments on this dataset, and the results demonstrate that our approach can generate accurate and harmonious human-object interactions, resulting in highly realistic and convincing compositions.

The main contributions of this work include:

1. We propose a new approach for interaction-aware human-object composition, named ***HO-Comp***, which focuses on seamlessly integrating a foreground object onto a human-centric background image while ensuring harmonious interactions and preserving the visual consistency of both the foreground object and the background person.

2. ***HOComp*** incorporates two innovative designs: *MLLMs-driven region-based pose guidance (MRPG)* for constraining human-object interaction via a *coarse-to-fine* strategy, and *detail-consistent appearance preservation (DCAP)* for maintaining consistent foreground/background appearances.

3. We introduce the *Interaction-aware Human-Object Composition (IHOC) dataset*, and conduct extensive experiments on this dataset to demonstrate the superiority of our method.

## 2 Related Works

**Image-guided Composition.** It aims to seamlessly integrate a user-provided foreground exemplar onto a designated region of a background image, sometimes with textual guidance. Existing methods either focus on appearance harmonization (*i.e.*, adjusting colors, shadows, and perspective) in order to integrate the foreground onto the background seamlessly [57, 98, 8, 60, 69, 61, 39, 79, 17, 7] or emphasize identity preservation by introducing dedicated modules to maintain the identity consistency of the object across scenes [9, 68, 84, 36, 99, 66]. However, these methods primarily refine the foreground and often fail to generate natural, realistic human gestures or poses in human-object interactions (HOIs). While DreamFuse [26] adjusts the foreground to adapt to the background context, it supports only limited hand actions and struggles with complex HOIs. Recent works [72, 74, 88, 80, 2, 45] propose unified frameworks to integrate multiple image generation/editing tasks. Similar to multi-modality methods [86, 52, 44], these approaches often unintentionally modify the background human and introduce inconsistencies in the foreground object.

**Multi-Concept Customization.** It aims to generate images that align with both the text prompt and user-specified concepts, facilitating the creation of personalized content. Tuning-based methods [34, 1, 71, 48, 47, 16, 41] typically incorporate new concepts into diffusion models by fine-tuning specific parameters, but each new concept requires a separate tuning process. Instead, training-based methods [87, 56, 103, 77, 33, 10, 42, 12, 40] train additional modules to extract the identity of a concept and inject it into the denoising network via attention layers. Training-free methods [14, 78, 96, 85] incorporate reference-aware attention mechanisms. These methods typically re-generate both the foreground object and background human, leading to inconsistent background human appearance.

**Human-Object Interaction (HOI) Generation.** It aims to synthesize images that depict plausible and coherent interactions between humans and objects. Recent diffusion-based methods generate HOI

---

[2]This interaction type is embedded in the text prompt. For example, "A woman is ***holding*** a hat", and "A kid is ***eating*** a donut."

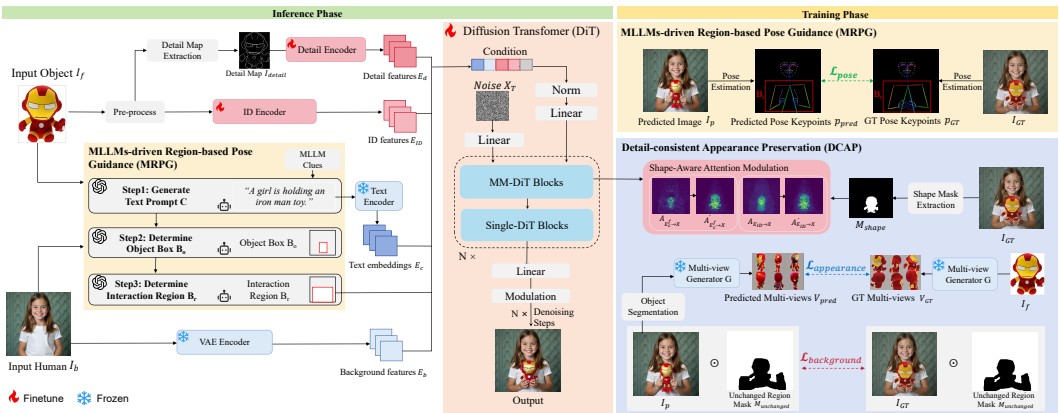

Figure 2: **Pipeline of *HOComp*.** Our method includes two core modules: MRPG for constraining human-object interaction and DCAP for maintaining appearance consistency. **Inference Phase (left):** MRPG uses MLLMs to generate a text prompt $C$, object box $B_o$ and interaction region $B_r$. Among these, $B_r$ and $C$ are encoded and, together with the object ID, detail features, and background features, are used to condition the DiT for final composition generation. **Training Phase (right):** MRPG constrains the interaction by applying a *pose-guided loss* $\mathcal{L}_{\text{pose}}$ with keypoint supervision. DCAP enforces appearance consistency via: (1) *shape-aware attention modulation* to adjust the attention maps to follow the object's shape prior $M_{\text{shape}}$; (2) a multi-view appearance loss $\mathcal{L}_{\text{appearance}}$ to semantically align synthesized and input foregrounds (multi-views); and (3) a background loss $\mathcal{L}_{\text{background}}$ to preserve original background details.

images/videos by introducing extra cues, such as bounding boxes [20, 30, 25] or pose structures [101, 38, 6], reference videos [89, 82], 3D priors [46] and in-context samples of similar interactions [27, 102, 28]. However, all these approaches require additional inputs during inference (*e.g.*, human poses or images describing the interaction). Some works [97, 91] adjust human hand poses during interactions, but this is often insufficient for complex scenarios. Other methods [63, 24, 54, 15, 93] employ relation-aware frameworks to improve HOI generation in subject-driven settings, yet they fail to preserve the background human appearance consistency. Concurrent works, DreamActor-H1 [76] and HunyuanVideo-HOMA [29], explore human interaction in the contexts of human-product demonstrations and animated human-object interactions. They incorporate additional modality guidance and exploit the strong multi-modal fusion capabilities of the DiT framework for video generation.

Several methods [13, 65, 75, 35, 94] focus on 3D human–object interaction generation, aiming to produce physically plausible 3D human–object interactions. These approaches typically generate 3D models or motion sequences conditioned on textual or semantic descriptions of the interaction. However, due to their reliance on 3D information as training references, the range of interaction types they can handle remains limited.

In summary, existing methods fall short in addressing the challenge of our interaction-aware human-object composition task, which requires the model to produce harmonious human-object interactions and consistent foreground/background appearances.

## 3   Method

Given a foreground object image $\mathbf{I_f}$ and a background image $\mathbf{I_b}$ containing a human subject, our goal is to synthesize a harmoniously composited image $\mathbf{I_p}$ that integrates the foreground object onto the human-centric background image. The composited image should exhibit harmonious interactions and maintain appearance consistency between the foreground object and the background human.

To achieve this objective, we propose ***HOComp***, an interaction-aware human-object composition framework, as illustrated in Fig. 2. Our framework includes two key components: *MLLM-driven Region-based Pose Guidance (MRPG)* and *Detail-Consistent Appearance Preservation (DCAP)*. MRPG leverages Multimodal Large Language Models (MLLMs) and human pose priors to constrain human-object interaction in a coarse-to-fine manner. DCAP preserves the shape and texture of the

foreground object while maintaining details of the background human, ensuring faithful and coherent appearance reproduction throughout the composited scene.

In the remainder of this section, we first introduce the preliminaries in Sec. 3.1. We then detail the design of MRPG in Sec. 3.2, followed by DCAP in Sec. 3.3. Finally, we describe our Interaction-aware Human-Object Composition (IHOC) dataset in Sec. 3.4.

## 3.1 Preliminary

**Diffusion Transformer** (DiT) is a transformer-based diffusion model for image synthesis. Given a noisy latent $\mathbf{z}_t$ at timestep $t$, it predicts the denoised output via $\hat{\mathbf{z}}_0 = \mathrm{DiT}(\mathbf{z}_t, t, c)$, where $c$ denotes a conditioning signal (*e.g.*, text embeddings or visual prompts). Owing to its scalability and strong generative capacity, DiT serves as a robust backbone for conditional image generation.

**Attention Manipulation** is a key strategy for improving semantic alignment and structural control in diffusion models through attention map editing, external signal injection, or modified attention weight computation. For a standard attention layer defined as $\mathbf{A} = \mathrm{softmax}(\mathbf{Q}\mathbf{K}^\top/\sqrt{d})\mathbf{V}$, manipulation introduces a structured bias or conditioning modulation: $\mathbf{A}' = \mathrm{softmax}((\mathbf{Q}\mathbf{K}^\top + \mathbf{M})/\sqrt{d})\mathbf{V}$, where $\mathbf{M} \in \mathbb{R}^{n \times n}$ encodes spatial priors or prompt-specific relevance (e.g., object masks).

## 3.2 MLLM-driven Region-based Pose Guidance (MRPG)

MRPG adopts a coarse-to-fine strategy to constrain the human-object interaction. At the coarse level, it leverages the reasoning capabilities of MLLMs to automatically identify suitable interaction type and corresponding interaction region through a multi-stage querying process. At the fine level, a *pose-guided loss* is introduced to impose fine-grained constraints on human poses within the interaction region, explicitly supervising the predicted image using human pose keypoints.

**Generating Interaction Regions and Types.** As illustrated in Fig. 2, we employ MLLMs (*e.g.,* GPT-4o) in a chain-of-thought, a step-by-step process to generate the interaction type (denoted as a text prompt $C$) and the interaction region (represented by a bounding-box $B_r$). While the interaction type specifies what interaction is to be performed by the background person on the foreground object (*e.g.*, holding), the interaction region specifies the location in the image that the interaction is to be performed. Specifically, we send the foreground object and the background image to the MLLM and query it in a three-stage approach: (1) With a set of initial prompts as the instruction guidance, we ask the MLLM to envision a plausible interaction type and return the interaction type in the form of a text prompt description $C$; (2) Conditioned on $C$, we ask the MLLM to further infer a potential region (*i.e.*, bounding box $B_o$) in the background image where the foreground object is to be placed; (3) We ask the MLLM to identify the interaction region $B_r$ by considering which human body parts are involved in the interaction. The generated interaction region $B_r$ is converted into a mask, encoded via a VAE [32], and used alongside text embeddings $E_c$ as conditioning inputs to the DiT model.

**Imposing Fine-grained Pose Guidance.** Considering the significant correlation between human-object interactions and body poses, we introduce a pose-guided loss $\mathcal{L}_{pose}$ to impose fine-grained constraints on poses within the interaction region. Let $\mathbf{p}_{\mathrm{GT}}^i$ and $\mathbf{p}_{\mathrm{pred}}^i$ represent the $i$-th keypoint detected by a pose estimator $\mathbf{G}_p$ from the ground-truth image $I_{\mathrm{GT}}$ and the predicted image $I_p$, respectively. The pose-guided loss $\mathcal{L}_p$ is formulated as:

$$\mathcal{L}_p = \frac{1}{n} \sum_{i \in B_r} \left\| \mathbf{p}_{\mathrm{GT}}^i - \mathbf{p}_{\mathrm{pred}}^i \right\|^2, \tag{1}$$

where $n$ denotes the number of pose keypoints located within the interaction region $B_r$, as illustrated in Fig. 2. This localized pose-guided loss explicitly directs the model's optimization efforts towards accurately capturing human poses involved in the interaction, rather than globally adjusting the entire human pose, thereby enhancing the realism and harmony of the generated interaction.

## 3.3 Detail-Consistent Appearance Preservation (DCAP)

To ensure fine-grained appearance consistency, for the **foreground**, we first extract identity and detail information as conditioning inputs for the DiT model. To enforce shape consistency, we introduce a *shape-aware attention modulation* mechanism to adjust the foreground-relevant attention maps in the

MM-DiT blocks, guiding the attention maps to align with the foreground object's shape prior better. For texture consistency, we propose a *multi-view appearance loss* to maintain semantic alignment across multiple viewpoints. For the **background**, we leverage an unchanged region mask to identify unaffected areas and impose a *background consistency loss* to preserve original background details.

**Foreground Object Identity and Detail Extraction.** We first preprocess the foreground object by removing the background and centering it. To capture the identity information, we then employ the DINOv2-based ID encoder [53], renowned for robust semantic representations, to extract the foreground ID features $E_{ID}$. As the resulting identity tokens have a coarse spatial resolution and therefore lack texture details, we extract a high-frequency detail map, $I_{detail}$, as an additional condition: $I_{detail} = I_{gray} - \mathrm{GaussianBlur}(I_{gray})$, where $I_{gray}$ is the grayscale foreground image. A lightweight detail encoder [9] processes $I_{detail}$ to extract detail features $E_d$, which are then fused with foreground ID features $E_{ID}$ to condition the DiT model.

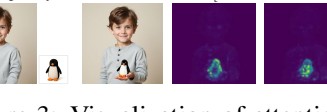

Input image & object    Result    $A_{E_c^f \to X}$    $A_{E_{ID} \to X}$

**Shape-aware Attention Modulation.** To enhance shape consistency, we modulate foreground-relevant attention maps in the MM-DiT blocks, encouraging the attention maps to align more precisely with the object's shape prior. This design is motivated by the observation that these attention maps highlight object shapes (see Fig. 3), indicating that the model is able to capture structural cues of the foreground objects.

Figure 3: Visualization of attention maps related to the foreground text embeddings $A_{\mathbf{E}_c^f \to \mathbf{X}}$ and the identity features $A_{\mathbf{E}_{ID} \to \mathbf{X}}$, both exhibiting strong alignment with object shape.

Specifically, we compute two foreground-relevant attention maps: one based on the foreground ID features $\mathbf{E}_{ID}$, and the other on the foreground text embeddings $\mathbf{E}_c^f$, with $\mathbf{X}$ denoting the target image tokens. Here, $\mathbf{E}_c^f$ are extracted from the full text embedding $\mathbf{E}_c$. For instance, if *"toy"* is annotated as a foreground object in the text prompt $C$ *"A boy is holding a toy"*, $\mathbf{E}_c^f$ is the sub-embedding aligned with *"toy"* from $\mathbf{E}_c$. The attention maps are computed as:

$$A_{\mathbf{E}_c^f \to \mathbf{X}} = \mathrm{softmax}\left(\frac{Q_{\mathbf{X}} K_{\mathbf{E}_c^f}^\top}{\sqrt{d}}\right), \quad A_{\mathbf{E}_{ID} \to \mathbf{X}} = \mathrm{softmax}\left(\frac{Q_{\mathbf{X}} K_{\mathbf{E}_{ID}}^\top}{\sqrt{d}}\right), \tag{2}$$

where $Q_{\mathbf{X}} \in \mathbb{R}^{N \times d}$ are queries from the target image tokens, and $K_{\mathbf{E}_c^f}, K_{\mathbf{E}_{ID}} \in \mathbb{R}^{M \times d}$ are keys projected from $\mathbf{E}_c^f$ and $\mathbf{E}_{ID}$, respectively.

To obtain the shape prior, as shown in Fig. 2, we extract a foreground object mask $M_{shape}$ from the ground-truth image. We aim to enhance the attention within the object region while suppressing distractions outside it. Considering that directly modifying the attention maps may potentially compromise the image quality of the pre-trained model [31], we adopt a residual-based modulation strategy over the extracted attention maps $A_{\mathbf{E}_c^f \to \mathbf{X}}$ and $A_{\mathbf{E}_{ID} \to \mathbf{X}}$ to incorporate shape priors while preserving the original attention distribution. The modulation is defined as:

$$A' = A + \alpha \cdot \left(M_{shape} \cdot (A_{max} - A) - (1 - M_{shape}) \cdot (A - A_{min})\right), \tag{3}$$

where $A \in \{A_{\mathbf{E}_c^f \to \mathbf{X}}, A_{\mathbf{E}_{ID} \to \mathbf{X}}\}$. $A_{max}$ and $A_{min}$ are the per-query maximum and minimum values computed row-wise. The scalar $\alpha \in \mathbb{R}^+$ controls the modulation strength. The modulated attention map is then integrated into the DiT model to encourage shape-aware feature learning.

**Multi-view Appearance Loss.** To address texture inconsistencies caused by changes in viewpoint during interactions, we encourage the predicted foreground object to maintain consistent semantic appearance with the ground truth across diverse views. Specifically, we synthesis multi-view images for both the predicted result and the input foreground, and measure their semantic similarity.

As shown in Fig. 2, we first segment the predicted foreground object from $\mathbf{I}_p$. Given the segmented output and the input foreground image $\mathbf{I}_f$, we apply a multi-view generator $G$ to synthesize $k$ views:

$$\mathbf{V}_{pred} = \{\mathbf{V}_{pred}^{(i)}\}_{i=1}^k = G(\mathrm{Segment}(\mathbf{I}_p)), \quad \mathbf{V}_{GT} = \{\mathbf{V}_{GT}^{(i)}\}_{i=1}^k = G(\mathbf{I}_f). \tag{4}$$

We then extract CLIP [58] features from each synthesized view: $\mathcal{F}_{pred}^{(i)} = \mathrm{CLIP}(\mathbf{V}_{pred}^{(i)}), \mathcal{F}_{GT}^{(i)} = \mathrm{CLIP}(\mathbf{V}_{GT}^{(i)})$. The multi-view appearance loss is then formulated as:

$$\mathcal{L}_{appearance} = \frac{1}{k} \sum_{i=1}^k \left(1 - \frac{\mathcal{F}_{pred}^{(i)} \cdot \mathcal{F}_{GT}^{(i)}}{\left\|\mathcal{F}_{pred}^{(i)}\right\| \left\|\mathcal{F}_{GT}^{(i)}\right\|}\right), \tag{5}$$

which encourages semantic alignment of the predicted object with the ground truth across multi-views.

**Background Consistency Loss.** To preserve the appearance of the background human during the process, we utilize an unchanged region mask $M_{\text{unchanged}}$, which is provided by our dataset and indicates the region that remains unaffected throughout the interaction. By constraining the generated image to match the ground-truth image in this unchanged region, we enforce consistency with the original background appearance. The background consistency loss $\mathcal{L}_b$ is defined as:

$$\mathcal{L}_{background} = \sum_{i \in I} M_{unchanged}^i \odot \left\| \mathbf{x}_{GT}^i - \mathbf{x}_{pred}^i \right\|^2, \tag{6}$$

where $\mathbf{x}_{GT}$ and $\mathbf{x}_{pred}$ denote the pixel values of the ground-truth image $\mathbf{I}_{GT}$ and of the predicted image $\mathbf{I}_p$, respectively.

**Overall Training Objective**. The model is optimized with the composite loss:

$$\mathcal{L}_{\text{total}} = \mathcal{L}_{\text{denoising}} + \alpha_1 \mathcal{L}_p + \alpha_2 \mathcal{L}_b + \alpha_3 \mathcal{L}_a, \tag{7}$$

where $\mathcal{L}_{\text{denoising}}$ is the standard denoising loss. $\mathcal{L}_p, \mathcal{L}_b, \mathcal{L}_a$ are the pose-guided, background consistency, and multi-view appearance losses. $\alpha_1, \alpha_2, \alpha_3$ are the coefficients of the corresponding loss terms.

### 3.4 Dataset Preparation

We introduce the *Interaction-aware Human-Object Composition (IHOC)* dataset to address the lack of paired pre- and post-interaction data crucial for modeling realistic and coherent human-object compositions. IHOC includes six components: (1) *background human images* (without the object); (2) *foreground object images*; (3) *composited images* with harmonious interactions and consistent appearances; (4) *text prompts* describing the interaction type; (5) *interaction regions*; and (6) *unchanged region masks* to indicate unaffected background areas.

Our dataset is constructed through the following stages: ❶ **Composited Images:** To enhance data diversity, we adopt the 117 human-object interaction types defined by HICO-DET [5] and include both real and synthetic samples. For real data, we manually select 50 images per type (5,850 total) from HICO-DET. To ensure the quality of our dataset and to reduce bias, we exclude images that (1) contain multiple persons, (2) lack clearly visible persons (*e.g.*, only a hand is shown), or (3) have large parts of the foreground objects occluded or not visible (*e.g.*, only one wheel of a bicycle is visible), making it difficult to identify them. The final selection emphasizes diversity in object type, scale, and human pose across diverse scenes. For synthetic data, we use GPT-4o to generate 50 prompts per type and synthesize 5,850 images using FLUX.1 [dev] [3]. These synthetic samples help complement the real data by introducing a wider range of human appearances, poses, viewpoints, and visual styles (*e.g.*, cartoon, sketches). In total, we have collected 11,700 composited interaction examples. ❷**Foreground Object Images:** Foreground objects are segmented from the composite images using SAM [59]. To address occlusions caused by human-object interactions, we use GPT-4o to infer and complete missing regions, producing plausible and visually consistent object appearances. ❸ **Background Human Images & Unchanged Region Masks:** We manually inpaint composite images using FLUX.1 Fill [dev] [4] to remove interacting objects and recover plausible human poses without the interactions. An inpainting mask denotes an interaction-altered region; its inverse produces the unchanged region mask, highlighting the area unaffected by the interaction. ❹ **Text Prompts & Interaction Regions**: For real images, we use GPT-4o to generate text prompts. For synthetic images, we reuse the generation prompts. In addition, we use GPT-4o to annotate each prompt with foreground object tokens, indicating which words correspond to the foreground objects. The interaction regions are derived by inverting the unchanged region masks. More information on our dataset, including statistics and visualizations, can be found in Sec. B of the Appendix.

## 4   Experiments

**Implementation Details.** We adopt FLUX.1 [dev] [3] as the base model and fine-tune it using LoRA [23] with rank 16, applied to the attention layers. All training images are resized to $512 \times 512$ resolution. The model is trained for 10,000 steps with a batch size of 2, using AdamW and a learning rate of 1e-5. Training takes approximately 20 hours on $2 \times$A100 GPUs. We employ DWPose [95] for pose estimation, Zero123+ [64] for multi-view generation and GPT-4o[52] as MLLM in MRPG.

Table 1: Quantitative comparison of our method with nine SOTA methods. The user study reports the averaged rank (lower is better) of nine methods in image quality (IQ), interaction harmonization (IH), and appearance preservation (AP). The best and second-best results are shown in **bold** and underlined, respectively. Training or tuning-based methods without released training codes are marked with a †.

| Category | Metrics | AnyDoor [9] | PbE [92] | FreeComp. [11] | FreeCustom [14] | PrimeComp. [79] | OmniGen [88] | GenArt. [80] | UniCom. [74] | GPT-4o† [52] | Ours |
|---|---|---|---|---|---|---|---|---|---|---|---|
| Automatic | FID ↓ | 18.57 | 15.91 | 22.55 | 18.57 | 17.48 | 12.13 | 14.52 | 11.55 | 9.98 | **9.27** |
| | CLIP-Score ↑ | 27.65 | 29.03 | 27.56 | 28.43 | 28.31 | 29.77 | 29.11 | 29.28 | 29.35 | **30.29** |
| | HOI-Score ↑ | 25.69 | 38.71 | 22.81 | 45.72 | 32.66 | 62.33 | 51.83 | 58.91 | 75.22 | **87.39** |
| | DINO-Score ↑ | 58.83 | 54.83 | 44.67 | 42.02 | 48.12 | 43.92 | 53.96 | 51.02 | 65.23 | **78.21** |
| | SSIM(BG) ↑ | 90.71 | 88.72 | 86.65 | 43.22 | 85.22 | 82.08 | 57.83 | 88.24 | 47.22 | **96.57** |
| User study | IQ ↓ | 9.72 | 7.47 | 8.20 | 9.13 | 3.23 | 2.63 | 6.22 | 3.93 | 3.10 | **1.37** |
| | IH ↓ | 8.18 | 8.23 | 8.46 | 6.72 | 6.68 | 5.23 | 4.88 | 2.87 | 2.61 | **1.14** |
| | AP ↓ | 2.84 | 5.41 | 6.84 | 7.33 | 6.07 | 4.73 | 6.54 | 8.26 | 5.87 | **1.11** |

(a) Input image & object   (b) GPT-4o[52]   (c) GenArt.[80]   (d) OmniGen[88]   (e) AnyDoor[9]   (f) PbE[92]   (g) UniCom.[74]   (h) Ours

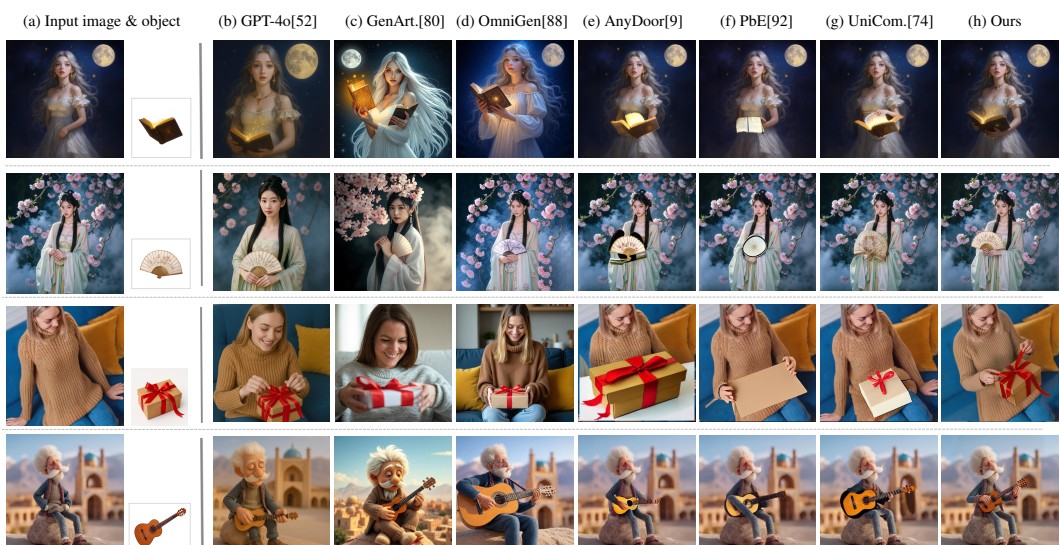

Figure 4: Qualitative comparison with six top performing SOTA methods from Table 16. The prompts for the above four examples are: "A girl is reading a magic book", "A woman is holding an ornate folding fan", "A woman is opening a gift box", and "A puppet-style old man is playing a guitar".

**Evaluation Metrics**. We use **FID** [19] to assess the overall quality of the generated images, where a lower score indicates a better alignment with real images. To evaluate how well a generated image depicts the specified human-object interaction (*i.e.*, HOI Alignment), we compute the **HOI-Score** using a pre-trained HOI detector (*e.g.*, UPT [100]), which measures the accuracy of the interaction in the generated image. Additionally, we employ the **CLIP-Score** [18] to evaluate the global semantic alignment between the generated image and the text prompt. Subsequently, we use the **DINO-Score** to assess how well the foreground object appearance is preserved, where a higher score indicates a better appearance consistency to the input foreground object. Finally, background consistency is evaluated by computing the Structural Similarity Index (**SSIM**) [81] over the area outside the interaction region, where a higher SSIM(BG) score indicates a better retention of the original background.

**Benchmark.** We introduce a new benchmark, **HOIBench**, to evaluate the quality of the human-object interaction task. We begin by collecting 30 images, each with a human person, from the internet. The humans in these images cover diverse appearances, including different poses and clothes. Half of these images feature the upper body, while the other half depict the full body. To ensure a broad range of interaction types, we adopt the 117 interaction types defined in the HICO-DET [22]. We prompt GPT-4o with each type to infer a plausible foreground object (*e.g.*, *playing → guitar*). A concise textual description of each object is then used to retrieve a representative image from the internet, yielding 117 interaction–foreground image pairs. Finally, for each human image, we randomly sample 20 interaction-object pairs from the generated set, producing a total of 600 human-object interaction instances (20 interactions × 30 human images) for evaluation.

Table 2: Ablation study on removing one of the key components from our full model (left table) and adding one of the key components to our base model (right table). $\mathcal{L}_p$, $\mathcal{L}_b$, $\mathcal{L}_a$, and SAAM denote the pose-guided loss, background consistency loss, multi-view appearance loss, and shape-aware attention modulation, respectively. Best performances are marked in **bold**.

| $\mathcal{L}_p$ | $\mathcal{L}_b$ | $\mathcal{L}_a$ | SAAM | FID↓ | CLIP↑ | HOI↑ | DINO↑ | SSIM(BG)↑ |
|---|---|---|---|---|---|---|---|---|
|  | ✓ | ✓ | ✓ | 14.24 | 28.05 | 34.42 | 69.32 | 94.91 |
| ✓ |  | ✓ | ✓ | 15.45 | 28.42 | 54.47 | 59.72 | 58.49 |
| ✓ | ✓ |  | ✓ | 13.31 | 29.37 | 67.32 | 46.12 | 95.11 |
| ✓ | ✓ | ✓ |  | 12.48 | 29.10 | 75.23 | 66.52 | 95.28 |
| ✓ | ✓ | ✓ | ✓ | **9.27** | **30.29** | **87.39** | **78.21** | **96.57** |

| $\mathcal{L}_p$ | $\mathcal{L}_b$ | $\mathcal{L}_a$ | SAAM | FID↓ | CLIP↑ | HOI↑ | DINO↑ | SSIM(BG)↑ |
|---|---|---|---|---|---|---|---|---|
|  |  |  |  | 21.25 | 26.14 | 26.76 | 22.19 | 34.91 |
| ✓ |  |  |  | 15.80 | 26.42 | 47.32 | 30.21 | 53.11 |
|  | ✓ |  |  | 14.72 | 26.83 | 30.08 | 33.54 | 93.29 |
|  |  | ✓ |  | 16.02 | 26.71 | 31.09 | 55.81 | 54.29 |
|  |  |  | ✓ | 16.21 | 26.51 | 29.85 | 42.53 | 57.32 |

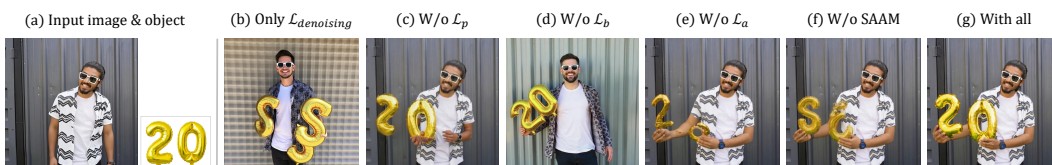

|(a) Input image & object|(b) Only $\mathcal{L}_{denoising}$|(c) W/o $\mathcal{L}_p$|(d) W/o $\mathcal{L}_b$|(e) W/o $\mathcal{L}_a$|(f) W/o SAAM|(g) With all|

Figure 5: Visual comparison of the ablation study in Table 2.

## 4.1 Comparison with State-of-the-Art Methods

We compare ***HOComp*** with 9 SOTA methods: AnyDoor [9], Paint by Example [92], FreeCompose [11], FreeCustom [14], OmniGen [88], GenArtist [80], PrimeComposer [79], UniCombine [74] and GPT-4o [52]. All methods with public training code are retrained or fine-tuned on our dataset.

**Quantitative Comparison.** Table 16 compares the performances of our method against the nine existing methods. The results in the top part of the table show that our method consistently outperforms all these baselines across all evaluation metrics. Specifically, it achieves the highest HOI-Score (87.39), surpassing GPT-4o by 12.17 and OmniGen by 25.06, underscoring its strong ability to model accurate and coherent human–object interactions. In terms of visual consistency, our method achieves the lowest FID (9.27) and the highest CLIP-Score (30.29), demonstrating superior realism and semantic alignment ability. Our DINO-Score (78.21) significantly outperforms AnyDoor by 19.38 and GPT-4o by 13.0, indicating improved foreground appearance consistency. Further, our model produces the most consistent background details with the highest SSIM(BG) score (96.57), outperforming AnyDoor by 5.86.

**Qualitative Comparison.** Fig. 4 visually compares the results of our method and those of the six top-performing methods from Table 16. Rows 3-4 of Fig. 4(b) show that although GPT-4o can synthesize plausible human–object interactions, it fails to maintain foreground appearance consistency. Meanwhile, its generated backgrounds exhibit substantial variations, as shown in rows 1-3 of Fig. 4(b). Similar to GPT-4o, GenArtist and OmniGen also suffer from foreground–background inconsistency. In addition, methods in Fig. 4(e-g) produce suboptimal or implausible hand poses. In contrast, our method effectively constrains the generated human poses as well as the shapes/textures of the foreground objects. As a result, the images produced by our method exhibit superior appearance consistency with harmonious human-object interactions.

**User Study.** We have also conducted a user study to compare our method with all 9 existing methods. We recruit a total of 75 student participants for the subjective assessment. Each participant is presented with 10 sets of cases, where each set contains an input human image, a foreground object, a text prompt to describe the interaction, and ten randomly shuffled results from ***HOComp*** and the 9 competing methods. Participants rank the images based on three criteria: image quality (IQ), interaction harmonization (IH), and appearance preservation (AP). We collect ranking scores from all participants and compute the average ranking for each of the three aspects, as shown in the bottom part of Table 16. These results show that our approach ranks first in all three aspects: image quality (1.37), interaction harmonization (1.14), and appearance preservation (1.11), highlighting it being the most preferred method by all participants.

### 4.2 Ablation Study

**Component Analysis.** We conduct an ablation study on *HOComp* by systematically removing one key component from our full model (Table 2 (left)) or by adding one key component to our base model (Table 2 (right)). Fig. 5 visualizes some results of the ablation study. Based on these results, we can draw six key conclusions: ❶ Pose constraint ($\mathcal{L}_p$) is essential for ensuring proper human pose generation during interactions. When removed, the result in Fig. 5(c) exhibits a distorted and incongruous interaction, leading to the lowest CLIP and HOI scores shown in row 1 of Table 2 (left). Its absence also lowers the SSIM(BG) score from 96.57 to 94.91, showing a mild but noticeable loss of background consistency. ❷ Background consistency loss ($\mathcal{L}_b$) helps prevent unintended modifications of non-interaction region of the background image. Without it, the person as well as the background scene may undergo significant changes (Fig. 5(d)), resulting in the worst FID score shown in row 2 of Table 2 (left). As a result, the SSIM(BG) score plummets to 58.49, the largest drop among all settings, causing the most severe background degradation. ❸ Multi-view appearance loss ($\mathcal{L}_a$) ensures consistency in the texture/appearance of the foreground object in the generated image. Removing it leads to noticeable color and texture shifts of the object (*e.g.*, the balloons in Fig. 5(e)) and the lowest DINO score shown in row 3 of Table 2 (left). ❹ Shape-aware attention modulation (SAAM) plays a crucial role in preserving object shape consistency. As shown in row 4 of Table 2 (left), removing SAAM leads to inconsistent shape transformations and appearance variations, with the DINO score dropping significantly from 78.21 to 66.52. ❺ Finally, by integrating all key components, our proposed method achieves the best performance, as shown in row 5 of Table 2 (left). ❻ Table 2 (right) shows that each component individually enhances a specific aspect of the model. $\mathcal{L}_p$ helps improve interaction quality, as reflected in higher HOI and CLIP scores. $\mathcal{L}_b$ improves background consistency, evident from the SSIM(BG) score. $\mathcal{L}_a$ and SAAM help maintain foreground appearance consistency, leading to improved DINO performances.

## 5 Conclusion

In this paper, we have presented *HOComp*, a framework for interaction-aware human-object composition. It leverages MLLM-driven region-based pose guidance (MRPG) for constrained human-object interaction, and detail-consistent appearance preservation (DCAP) for maintaining appearance consistency. To support *HOComp* training, we have also introduced the Interaction-aware Human-Object Composition (IHOC) dataset. Extensive experiments demonstrate that *HOComp* outperforms existing methods in quantitative, qualitative, and subjective evaluations.

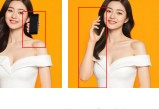

Figure 6: An example failure case of *HOComp*. The red boxes indicate the interaction regions.

*HOComp* does have limitations. Although MLLMs correctly identify the interaction region in 91.33% of the samples in our benchmark, HOIBench, incorrect predictions may still affect the quality of the generated interactions, as shown in Fig. 6. As a future work, we would like to consider incorporating human pose priors into predicting the interaction region.

## Acknowledgements

This work is supported in part by Ant Group and General Program of National Natural Science Foundation of China (NSFC, No. 6207071897).

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

# HOComp: Interaction-Aware Human-Object Composition

## Appendix

## A    Overview

In this appendix, we provide additional implementation details, ablation analyses, and extended evaluations to further support and expand upon the findings presented in the main paper.

Specifically, we address the following key aspects in our appendix: (1) Presenting detailed statistical analyses and the construction procedure of our *IHOC* dataset (Sec. B); (2) Offering additional clarifications on our approach, including experimental configurations and supplementary ablation analyses (Sec. C- H); (3) Presenting additional experiments to validate our method, including further comparisons with state-of-the-art approaches and more results of our method (Sec. I- K). (4) Discussing on ethical considerations, data governance procedures, and responsible use safeguards implemented in the development of our method. (Sec. L);

## B    Extended Details on *IHOC* dataset

### B.1    Dataset Construction

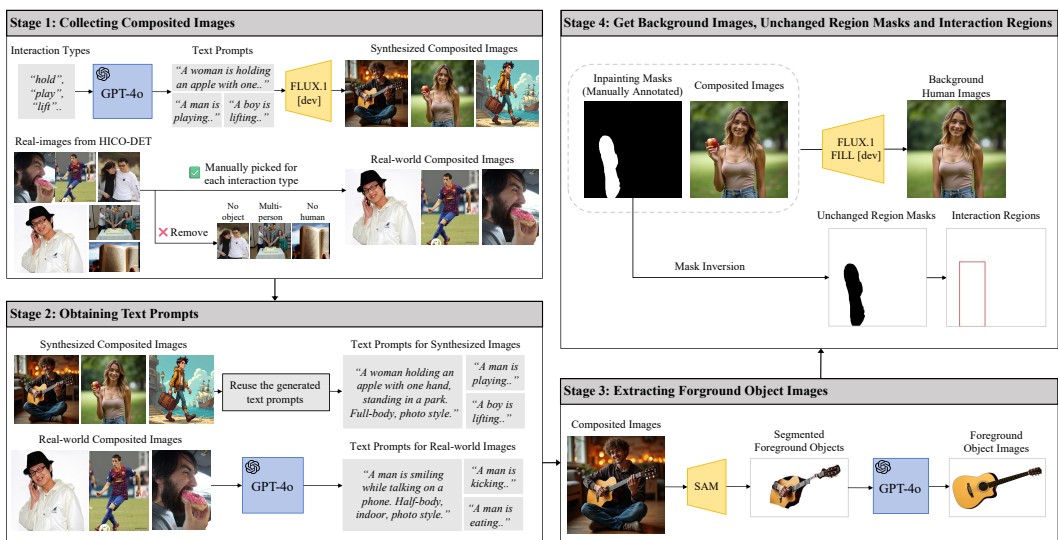

Figure 7: Overview of the construction process of our *Interaction-aware Human-Object Composition (IHOC) dataset*. It involves four stages: (1) collecting synthesized and real-world composited images, (2) obtaining corresponding text prompts, (3) extracting foreground object images, and (4) getting background human images, unchanged region masks, and interaction regions.

In Sec. 3.4 of the main paper, we briefly discuss our *Interaction-aware Human-Object Composition (IHOC) dataset*, which includes six components: (1) background human images (without the object); (2) foreground object images; (3) composited images with harmonious interactions and consistent appearances; (4) text prompts describing the interaction type; (5) interaction regions; and (6) unchanged region masks to indicate unaffected background areas. As shown in Fig. 7, our IHOC dataset construction comprises four stages.

**Stage 1: Collecting synthesized and real composited images.** To ensure data diversity, we adopt the 117 human-object interaction categories from HICO-DET[22], comprising both real and synthetic samples. For real images, we manually selected 50 images per category, resulting in a total of 5,850 from HICO-DET, excluding those that (1) contain multiple people, (2) lack clearly visible humans, or (3) lack clearly visible objects, which impair recognizability. The final set emphasizes diversity in

object type, scale, and human pose across scenes. For synthetic images, we use GPT-4o to generate 50 text prompts per category and synthesize 5,850 samples using FLUX.1 [dev][3]. These images complement the real data by introducing broader variations in human appearance, pose, viewpoint, and visual style (e.g., cartoons, sketches). In total, we collect 11,700 composited images.

**Stage 2: Generating text prompts.** For real images, we use GPT-4o to generate descriptive prompts. For synthetic images, we reuse the prompts originally used for generation.

**Stage 3: Extracting foreground objects.** We segment foreground objects from composited images using SAM [59]. To address occlusions caused by human-object interactions, GPT-4o infers and fills missing regions, producing complete and visually consistent objects.

**Stage 4: Getting background images, unchanged region masks, and interaction regions.** We manually annotate inpainting masks and use FLUX.1 FILL [dev] [4] to remove interacting objects and reconstruct plausible human poses without interactions. The inpainting masks define interaction-affected regions; their inverse yields the unchanged region masks. Interaction regions are computed by extracting the minimal bounding box of the interaction area within the unchanged region mask.

## B.2 Dataset Statistics

As shown in Fig. 8, our dataset consists of six components: (1) background human images (without the object); (2) foreground object images; (3) composited images with harmonious interactions and consistent appearances; (4) unchanged region masks to indicate unaffected background areas; (5) interaction regions and (6) text prompts describing the interaction type;

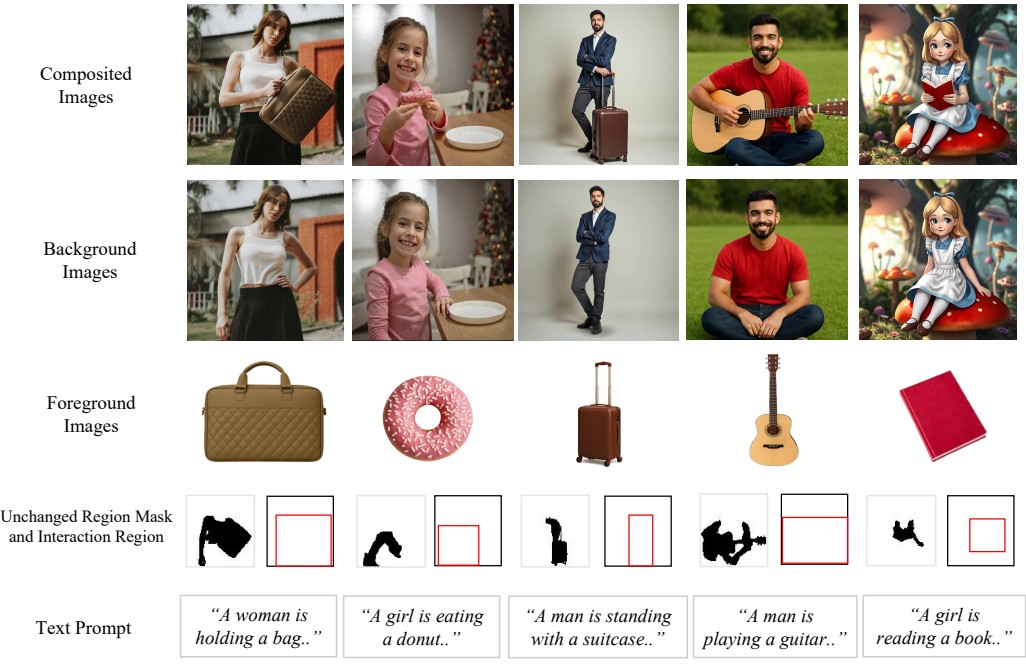

Figure 8: Visualization of our Interaction-aware Human-Object Composition (IHOC) Dataset.

Our dataset consists of 11,700 composited images, with half sourced from real-world data and the other half generated synthetically. Our dataset comprises a total of 117 types of interaction types and 342 distinct foreground object categories. To highlight the diversity of our dataset, we analyze its statistical properties across six dimensions, as illustrated in Fig. 9(a–f):

**(1) Human Viewpoint:** Our dataset includes four distinct human viewpoints, categorized by body visibility and camera angle: full-body frontal, full-body side, upper-body frontal, and upper-body side (see Fig. 9(a)). Upper-body frontal is the most common (42.4%), followed by full-body frontal (27.5%), upper-body side (15.7%), and full-body side (14.5%). This distribution is reasonable, as frontal views typically support a wider range of interaction types and are more frequently used in practice.

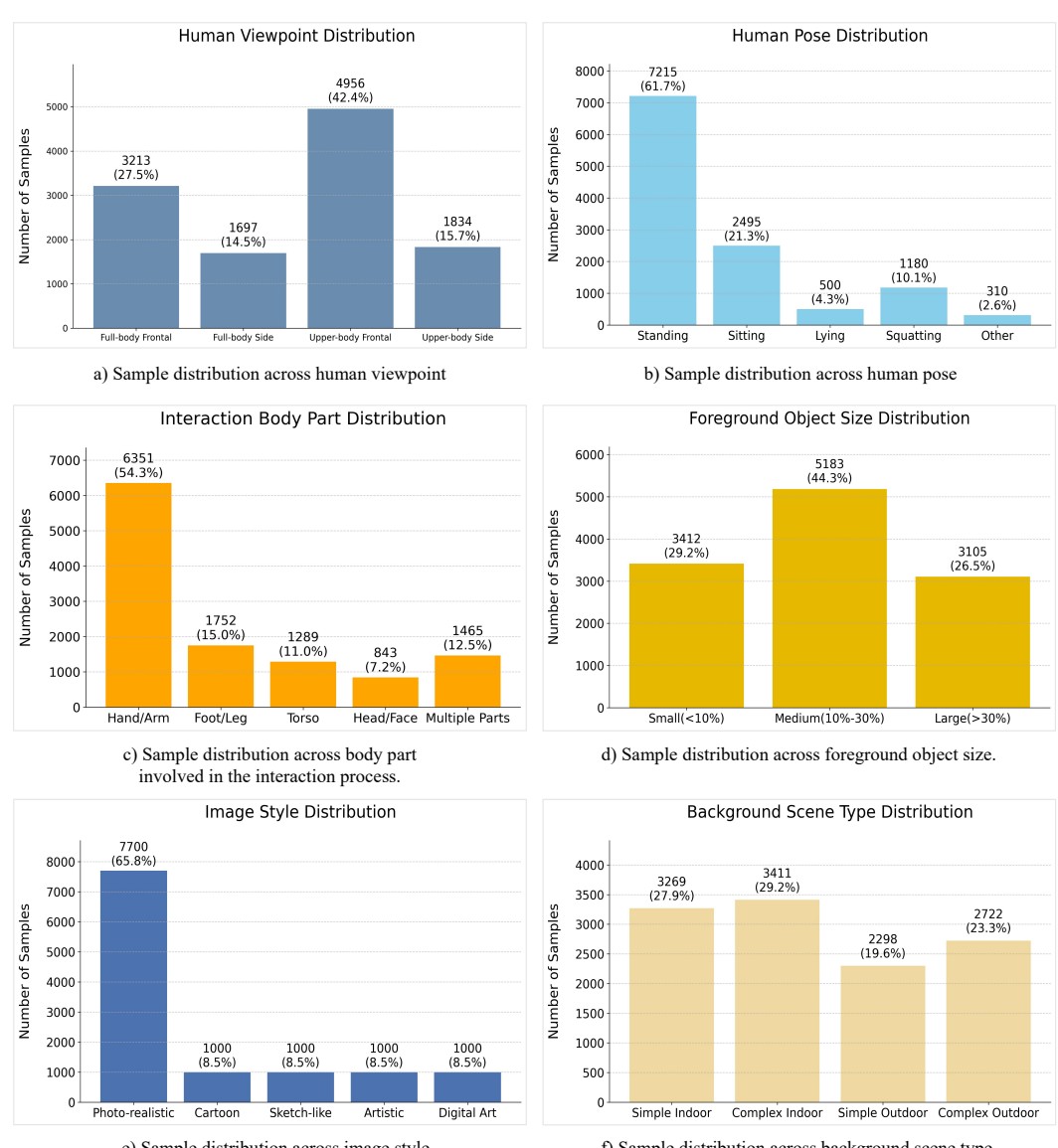

Figure 9: Statistical analysis of our *Interaction-aware Human-Object Composition (IHOC) dataset* across six dimensions: (a) human viewpoint, (b) human pose, (c) interaction body part, (d) foreground object size, (e) image style, and (f) background scene type. These statistics demonstrate the dataset's diversity in visual appearance, interaction types, and contextual complexity.

**(2) Human Pose:** Our dataset covers five major categories of human pose: standing, sitting, lying, squatting, and other (*e.g.*, jumping on a skateboard) (see Fig. 9(b)). Standing is the most prevalent (61.7%), followed by sitting (21.3%), squatting (10.1%), lying (4.3%), and other (2.6%). This distribution demonstrates that our dataset includes both common and less frequent poses.

**(3) Interaction Body Part:** We categorize the interactions in our dataset into five body regions based on which part of the body changes position before and after the interaction: hand/arm, foot/leg, torso, head/face, and multiple parts (see Fig. 9(c)). Hand/arm interactions are the most dominant (54.3%), other interactions involve foot/leg (15.0%), multiple parts (12.5%), torso (11.0%), and head/face (7.2%). This distribution highlights the diversity of interaction types and the involved body regions in our dataset.

**(4) Foreground Object Size:** Our dataset includes foreground objects of varying sizes. Based on the ratio of foreground object area to the entire image area, we classify them into three categories: small

(<10%), medium (10–30%), and large (>30%) (see Fig. 9(d)). Medium objects are the most common (44.3%), followed by small (29.2%) and large (26.5%). This distribution indicates that our dataset captures a diverse range of object sizes, which is essential for evaluating interaction robustness across different foreground scales.

**(5) Image Style:** Our dataset spans five distinct image styles: photo-realistic, cartoon, sketch-like, artistic, and digital art (see Fig. 9(e)). Photo-realistic images comprise the majority (65.8%), while the remaining styles each account for 8.5%. This diversity supports our method in handling images from different visual domains.

**(6) Background Scene Type:** Our dataset includes images with diverse background scenes, which we use GPT-4o to judge the complexity of background scene: simple indoor, complex indoor, simple outdoor, and complex outdoor (see Fig. 9(f)). The distribution is relatively balanced: complex indoor (29.2%), simple indoor (27.9%), complex outdoor (23.3%), and simple outdoor (19.6%), ensuring broad coverage across varied scene contexts.

## C    Generalizability of HOComp.

To address the generalizability concern of our model, we have evaluated ***HOComp*** under challenging and diverse conditions.

We first validated our method's performance under three extreme or special cases: (1) multi-subject and multi-person compositions, (2) extremely large interaction regions, and (3) interactions involving completely occluded target objects.

### 1. Multi-subject and multi-person compositions:

We have tested ***HOComp*** on inputs with multiple persons and objects, focusing on cases where multiple persons interact with several objects simultaneously. Since HOIBench lacks multi-persons data, we have constructed a new benchmark with two scenarios: (1) two persons with two objects and (2) three persons with three objects.

Test Set Construction: We have collected 40 diverse images (20 with two persons, 20 with three persons) from the internet, and generated 117 interaction–foreground image pairs following the same procedure of constructing HOIBench described in Sec. 4.

For **Two persons, two objects:** Each person in an image is randomly assigned a unique interaction–object pair, ensuring a wide variety of combinations. For each of the 20 two-person images, we repeat this assignment process 10 times with different pairs, resulting in 200 two-person, two-object test cases.

For **Three persons, three objects:** Similarly, each person is assigned a unique interaction–object pair, randomly sampled from the 117 pairs described above. This process is repeated for each three-person image, resulting in 200 three-person, three-object cases.

We have evaluated our model using the new test set of 400 cases (200/scenario) and compared its performance with the top three methods from the main paper. We compute the HOI-Score and DINO-Score for each individual person-object interaction pair, and then take the average of all these scores across all interaction pairs. As shown in the following table, our approach achieves the best results in both scenarios and on all metrics.

Table 3: Quantitative results of our method on multi-subject and multi-person compositions.

| Category | Method | FID↓ | CLIP-Score↑ | HOI-Score↑ | DINO-Score↑ | SSIM(BG)↑ |
|---|---|---|---|---|---|---|
| 2 persons & 2 objects | AnyDoor[9] | 22.38 | 26.86 | 23.12 | 55.46 | 83.19 |
| | OmniGen.[88] | 15.98 | 29.03 | 51.45 | 38.66 | 75.91 |
| | GPT-4o[52] | 11.13 | 29.18 | 73.12 | 63.17 | 25.50 |
| | **Ours** | **10.25** | **29.87** | **83.61** | **73.48** | **93.25** |
| 3 persons & 3 objects | AnyDoor[9] | 24.17 | 26.03 | 22.98 | 52.07 | 80.11 |
| | OmniGen.[88] | 18.25 | 28.14 | 49.01 | 35.85 | 72.14 |
| | GPT-4o[52] | 13.46 | 28.85 | 70.88 | 55.14 | 19.80 |
| | **Ours** | **10.43** | **29.76** | **82.98** | **73.09** | **91.52** |

### 2. Extremely large interaction regions:

We have also evaluated HOComp's performance on cases with exceptionally large interaction regions, such as those illustrated in Fig. 8 (e.g., "a man is playing a guitar," where the interaction covers over 70% of the image). We have identified 82 images in HOIBench where the interaction region occupies more than 70% of the total image area, and compared HOComp with previous methods on this subset. As shown in the following table, HOComp achieves substantially better results than other baselines across all metrics in large interaction region cases.

Table 4: Quantitative results of our method on extremely large interactions region cases.

| Method | FID↓ | CLIP-Score↑ | HOI-Score↑ | DINO-Score↑ | SSIM(BG)↑ |
|---|---|---|---|---|---|
| AnyDoor[9] | 21.80 | 26.93 | 22.96 | 44.62 | 85.78 |
| OmniGen.[88] | 15.31 | 29.14 | 55.41 | 38.45 | 72.83 |
| UniCom.[74] | 13.55 | 29.17 | 54.22 | 49.37 | 84.02 |
| GPT-4o[52] | 10.03 | 29.12 | 74.01 | 63.39 | 32.91 |
| **Ours** | **9.44** | **30.26** | **85.11** | **77.48** | **95.43** |

## 3. Interactions involving completely occluded target objects:

Some actions may cause the interacting object to become entirely occluded in the resulting image, e.g., food being **eaten** (fully inside the mouth) or an object **blocked** by a car. To evaluate the model's generalization in such cases, we have curated 50 test instances where the target object is fully occluded after the interaction. We append "the object is occluded because of the human action" to the MLLM-generated descriptions, forming a dedicated evaluation subset.

Standard metrics requiring object visibility (e.g., DINO-Score, HOI-Score) are inapplicable here. Instead, we manually assess the plausibility of the depicted interactions ("Occlusion Accuracy"), focusing on whether the generated images realistically reflect the intended occlusion. The results in the table below clearly demonstrate the superiority of our method over existing works.

Table 5: Quantitative results of our method on cases involving completely occluded target objects.

| Method | FID↓ | CLIP-Score↑ | Occlusion Accuracy↑ | SSIM(BG)↑ |
|---|---|---|---|---|
| AnyDoor[9] | 22.95 | 26.85 | 0.08 | 83.52 |
| OmniGen.[88] | 15.83 | 29.07 | 0.42 | 71.89 |
| UniCom.[74] | 15.58 | 29.06 | 0.28 | 83.62 |
| GPT-4o[52] | 10.52 | 29.31 | 0.84 | 25.28 |
| **Ours** | **9.82** | **29.87** | **0.86** | **94.35** |

**Extension to the User Study:** To further demonstrate the generalization of our method to diverse inputs, we have expanded the user study in the main paper from 10 to 50 input cases, covering a broader range of human poses, viewpoints, and object categories. 45 participants have completed the evaluation. As shown below, our method consistently outperforms all baselines across all metrics, indicating strong user preferences for our approach.

Table 6: Extension user study results of our method.

| Metrics | Any.[9] | PbE[92] | FreeC.[11] | FreeCu.[14] | Prime.[79] | Omni.[88] | GenArt.[80] | UniC.[74] | GPT-4o[52] | Ours |
|---|---|---|---|---|---|---|---|---|---|---|
| IQ↓ | 8.28 | 7.64 | 9.66 | 9.16 | 3.14 | 2.49 | 6.03 | 4.07 | 3.29 | **1.24** |
| IH↓ | 8.65 | 8.90 | 8.35 | 5.85 | 6.15 | 5.01 | 5.34 | 2.91 | 2.69 | **1.15** |
| AP↓ | 3.05 | 5.04 | 7.20 | 6.92 | 6.25 | 4.93 | 6.65 | 8.05 | 5.74 | **1.17** |

Our framework can also be adapted to different foundation models. In our original experiments, we used FLUX.1[dev] as the base DiT model. We additionally evaluated HOComp with FLUX.1 Kontext[dev][2] as the base model. As shown below, HOComp attains superior performance on FLUX.1 Kontext[dev], further demonstrating the adaptability of our approach across base model variants.

Table 7: Results on different foundation models.

| Base DiT Models | FID ↓ | CLIP-Score↑ | HOI-Score↑ | DINO-Score↑ | SSIM (BG)↑ |
|---|---|---|---|---|---|
| FLUX.1[dev][3] | 9.27 | 30.29 | 87.39 | 78.21 | 96.57 |
| FLUX.1 Kontext[dev][2] | **9.19** | **30.32** | **89.21** | **81.13** | **97.34** |

# D More Discussions on Limitations of Our Method

Our MRPG module adopts a coarse-to-fine strategy for constraining human-object interactions. At the coarse level, it leverages MLLMs to automatically identify suitable interaction types/regions via multi-stage querying. In some rare cases, MLLMs may fail to accurately predict the interaction regions. We provide a detailed analysis of these failure modes and potential solutions below.

In our HOIBench experiments, MLLMs (e.g., GPT-4o) identified reasonable interaction types in **100%** of samples. The corresponding interaction region was predicted correctly in **91.33%** of cases (548/600). For the remaining **8.67%** (52/600), two main failure modes are observed:

1. **Mismatch Between Interaction Regions and Types (6% of cases):** When multiple plausible interaction types exist, MLLMs may misassign interaction regions. For example, if the object is sunglasses and the interaction type is "hold", the model may incorrectly assign the region around the eyes (i.e., "wear" action). While the generated image depicts a person wearing sunglasses with harmonious interactions and consistent appearance, the predicted interaction type is inconsistent.

2. **Incorrect Interaction Region Size (2.67% of cases):** In these cases, the predicted interaction region does not fully cover the area, including the object and the human body part, for modification. As shown in Fig. 6, this can hinder generation of correct interactions.

To address these two issues, we explored the following solutions:

- **Additional Input Conditions:** To alleviate the mismatching issue, we can further incorporate human pose priors (from a pose estimator) as an additional prior to MLLMs. For GPT-4o, the region prediction accuracy increased from **91.33%** to **96.5%**. This improvement can be attributed to the introduction of explicit keypoint information, which provides precise localization of body parts such as the face and hands.

- **Training with Noisy Data:** To mitigate the impact of incorrect interaction region sizes, we introduce noisy data during training. We first generated 1,000 accurate input data samples. In accordance with the observed error rate, we then moved the bounding boxes of 700 samples to create mismatches with the interaction types (e.g., moving the bounding box of a "soccer ball" with a "kick" action to cover the hand, which may correspond to a "hold" action). Additionally, we reduced the bounding boxes of 300 samples so that they no longer fully covered the interaction object or the required body movement. We fine-tuned the previously pretrained HOComp model for 3,000 steps using these noisy samples alongside the original IHOC dataset. After retraining, we re-evaluated our method using the 52 failure cases. As shown in the following table, training with noisy data substantially improved the model's performance even when MLLMs mispredicted the interaction region.

Table 8: Results of our method using different training strategies.

| Training Strategy | FID↓ | CLIP-Score↑ | HOI-Score↑ | DINO-Score↑ | SSIM (BG)↑ |
|---|---|---|---|---|---|
| Without Noisy Data | 11.58 | 29.55 | 38.62 | 70.02 | 62.01 |
| With Noisy Data | **10.03** | **29.82** | **77.89** | **74.29** | **88.93** |

# E Effectiveness of Residual-based Modulation Strategy

As discussed in Sec. 3.3 of the main paper, our shape-aware attention modulation employs a residual-based strategy to adjust the attention maps. This design is motivated by the concern that directly

modifying attention maps may degrade the visual quality of the generated images, as suggested by previous work [31].

We define our modulation as:

$$A' = A + \alpha \cdot (M_{\text{shape}} \cdot (A_{\max} - A) - (1 - M_{\text{shape}}) \cdot (A - A_{\min}))$$

where $A$ is the original attention map, $M_{\text{shape}}$ is the ground-truth shape mask, $\alpha$ is a modulation strength, $A_{\max}$ and $A_{\min}$ denote the maximum and minimum attention values per query. The terms $(A_{\max} - A)$ and $(A - A_{\min})$ serve as residuals, which helps constrain the modulation within the original attention range. This ensures that the updated attention map $A'$ does not deviate excessively, thereby preserving the pretrained model's attention distribution. For comparison, we also evaluate a naive modulation strategy without residual constraints, formulated as:

$$A' = A + \alpha \cdot (M_{\text{shape}} - (1 - M_{\text{shape}}))$$

We conduct an ablation study on the HOIBench to compare the effectiveness of the residual-based strategy versus the non-residual version. As shown in Fig. 10 and Table. 9, removing the residual leads to a notable drop in FID and DINO scores, indicating degraded image quality and reduced consistency of the generated foreground objects. Other metrics also show minor decreases. Visually, the generated shapes deviate more from the input guidance, confirming the importance of the residual design.

Table 9: Ablation study on attention modulation strategies.

| Modulation Strategy | FID ↓ | CLIP ↑ | HOI ↑ | DINO ↑ | SSIM(BG) ↑ |
|---|---|---|---|---|---|
| Non-residual Strategy | 10.89 | 30.07 | 84.32 | 69.72 | 95.58 |
| Residual Strategy | **9.27** | **30.29** | **87.39** | **78.21** | **96.57** |

|  (a)Input image & object | (b)Residual Strategy | (c)Non-residual Strategy |

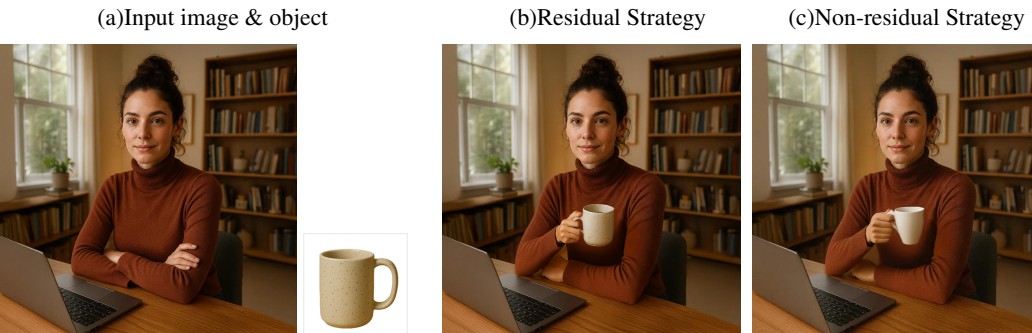

Figure 10: Visual results of ablation study on attention modulation strategies in Table 9.

## F   Effect of Coefficients

We evaluate the impact of four coefficients in the overall training loss and the shape-aware attention modulation on HOIBench. Specifically, $\alpha_1$, $\alpha_2$, and $\alpha_3$ are the coefficients of the pose-guided loss, background consistency loss, and multi-view appearance loss, respectively. $\alpha$ denotes the modulation strength used in the shape-aware attention modulation.

As shown in Table. 10. ❶Increasing $\alpha_1$ from 1 to 1.5 (Rows 1 vs. 2) improves HOI score (87.39 → 88.01) and CLIP score (30.29 → 30.31), indicating better pose alignment. However, this comes at the cost of image quality and consistency, with FID increasing (9.27 → 10.65), and both DINO and SSIM(BG) decreasing (78.21 → 73.32, 96.57 → 94.33). ❷Raising $\alpha_2$ from 0.5 to 1.0 (Rows 1 vs. 3) improves SSIM(BG) (96.57 → 96.92), reflecting better background preservation, but significantly degrades other metrics including FID, CLIP, HOI, and DINO—suggesting that excessive emphasis on background stability impairs semantic and visual coherence. ❸Increasing $\alpha_3$ from 0.8 to 1.0 (Rows 1 vs. 4) slightly improves DINO (78.21 → 78.58), indicating enhanced shape alignment, but at the cost of higher FID (12.92) and lower SSIM(BG) (94.88), showing a trade-off between appearance consistency and image quality. ❹Finally, increasing modulation strength $\alpha$ from 1.0 to 1.5 (Rows

1 vs. 5) causes moderate declines in FID (9.27 → 10.87), DINO (78.21 → 77.63), and SSIM(BG) (96.57 → 95.48), this effect may arise due to the destabilization of the pretrained attention distribution caused by excessively aggressive attention modulation.

Table 10: Quantitative comparison of different coefficient combinations. $\alpha_1$, $\alpha_2$, and $\alpha_3$ are the coefficients of the pose-guided loss, background consistency loss, and multi-view appearance loss, respectively. $\alpha$ denotes the modulation strength used in the shape-aware attention modulation.

| Coefficients ($\alpha_1$, $\alpha_2$, $\alpha_3$, $\alpha$) | FID ↓ | CLIP ↑ | HOI ↑ | DINO ↑ | SSIM(BG) ↑ |
|---|---|---|---|---|---|
| $\alpha_1$=1, $\alpha_2$=0.5, $\alpha_3$=0.8, $\alpha$=1 | **9.27** | **30.29** | **87.39** | **78.21** | **96.57** |
| $\alpha_1$=1.5, $\alpha_2$=0.5, $\alpha_3$=0.8, $\alpha$=1 | 10.65 | 30.31 | 88.01 | 73.32 | 94.33 |
| $\alpha_1$=1, $\alpha_2$=1, $\alpha_3$=0.8, $\alpha$=1 | 11.29 | 29.88 | 82.16 | 74.10 | 96.92 |
| $\alpha_1$=1, $\alpha_2$=0.5, $\alpha_3$=1, $\alpha$=1 | 12.92 | 29.71 | 85.75 | 78.58 | 94.88 |
| $\alpha_1$=1, $\alpha_2$=0.5, $\alpha_3$=0.8, $\alpha$=1.5 | 10.87 | 30.25 | 86.11 | 77.63 | 95.48 |

## G   Extended Details on Using MLLMs to Identify Interaction Types and Regions

In Sec. 3.2 of the main paper, we briefly described the use of MLLMs to infer interaction types and interaction regions via multi-turn querying. Here, we detail the full process.

Given a background human image $I_b$ and a foreground object image $I_f$, we iteratively use an MLLM to extract: (1) a text prompt $C$ describing the interaction, (2) the object bounding box $B_o$, and (3) the interaction region on the human $B_r$. The multi-turn procedure proceeds as follows:

1. **Interaction Prompt Generation.** The MLLM is queried with $I_f$ and $I_b$ using the instruction: *"Please analyze and describe a suitable type of interaction between them and generate a simple prompt for this interaction."* The model outputs a text prompt $C$ describing the interaction type.

2. **Object Box Prediction.** Using $I_f$, $I_b$, and $C$, we query the MLLM with: *"Please describe the position of the foreground object and give bounding box coordinates so that it aligns with the specified interaction."* The model returns the object bounding box $B_o$.

3. **Interaction Region Prediction.** Given $I_f$, $I_b$, $C$, and $B_o$, we ask: *"Based on the images and interaction prompt, and assuming the object is at $B_o$, identify the regions on the person that would be affected during the interaction and return their bounding box."* The MLLM then predicts the interaction region box $B_r$.

## H   Additional Ablation studies

### H.1   Multi-View Generators and View Numbers

We evaluate the impact of the number of views used in the multi-view appearance loss (Fig. 11, Table. 11 (left)). Using only a single view leads to noticeable inconsistencies in object appearance. As the number of views increases, performance improves steadily across all metrics, confirming the value of richer multi-view supervision.

We further evaluate different multi-view generation methods (Fig. 12, Table. 11 (right)). Without multi-view supervision, the model fails to maintain appearance consistency under significant viewpoint changes. Incorporating multiple generated views into the CLIP loss enhances coherence across varying poses and backgrounds. Among the methods, Zero123+[55] achieves the best results, while SV3D[73] and ViewDiff [21] also outperform the no multi-view baseline, underscoring the importance of high-fidelity multi-view supervision.

### H.2   LoRA Ranks

Table 12 presents the results of varying the LoRA rank (8, 16, 32, 64) across five evaluation metrics. Rank 16 consistently achieves the best overall performance, yielding the lowest FID (9.27) and the

Table 12: Ablation study on LoRA Ranks

| Rank | FID ↓ | CLIP ↑ | HOI ↑ | DINO ↑ | SSIM(BG) ↑ |
|---|---|---|---|---|---|
| 8 | 9.51 | 29.98 | 84.32 | 74.72 | 96.12 |
| **16** | **9.27** | **30.29** | **87.39** | **78.21** | **96.57** |
| 32 | 9.84 | 30.24 | 86.68 | 77.26 | 96.15 |
| 64 | 9.33 | 30.27 | 85.49 | 77.12 | 96.04 |

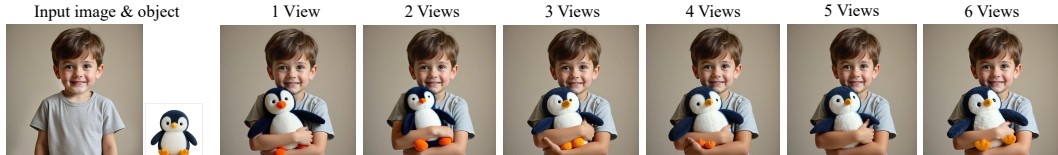

| Input image & object | 1 View | 2 Views | 3 Views | 4 Views | 5 Views | 6 Views |

Figure 11: Visual results of ablation study on view numbers used in multi-view appearance loss.

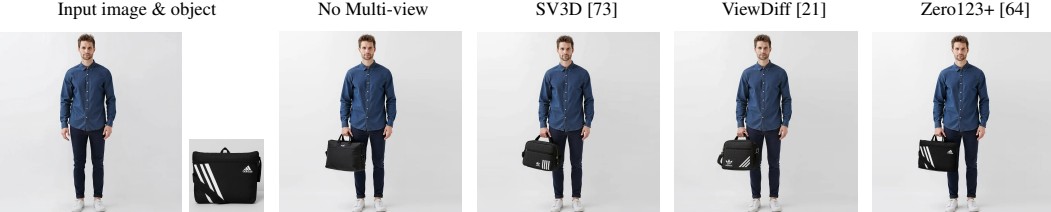

| Input image & object | No Multi-view | SV3D [73] | ViewDiff [21] | Zero123+ [64] |

Figure 12: Visual results of ablation study on multi-view generators.

highest scores in CLIP (30.29), HOI (87.39), DINO (78.21), and SSIM(BG) (96.57). When the rank is too low (e.g., 8), the model underperforms across all metrics, indicating insufficient capacity to model human-object interactions and maintain consistent appearances. However, higher ranks (32, 64) yield marginal or no improvements (*e.g.*, DINO drops to 77.26 and 77.12), suggesting possible overfitting.

### H.3 ID Encoder Backbone

As discussed in Sec. 3.3 of the main paper, we adopt DINOv2 as the backbone for extracting object identity features. Here, we conduct an ablation study comparing different backbones: VAE [51], CLIP [62], and DINOv2 [43]. To ensure a fair evaluation, we additionally report CLIP-I [58], which measures the CLIP similarity between the synthesized foreground object and the input foreground object.

As shown in Table. 13, DINOv2 consistently outperforms other ID encoder backbones across all evaluated metrics. As shown in Fig. 13, using DINOv2 as the ID encoder backbone yields the most consistent foreground object.

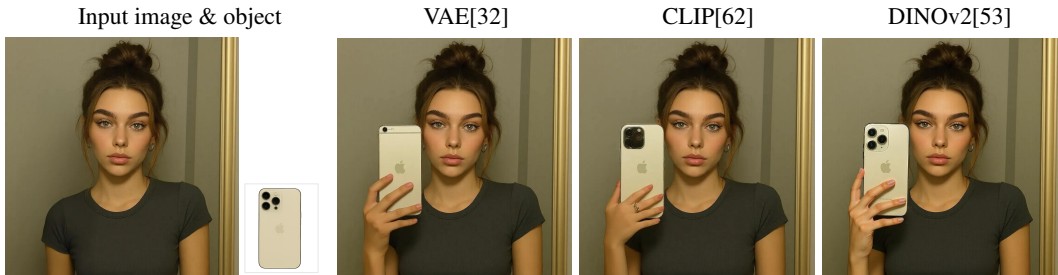

| Input image & object | VAE[32] | CLIP[62] | DINOv2[53] |

Figure 13: Ablation study on different backbones for foreground ID encoders.

### H.4 Guidance Scale

To study the impact of the guidance scale on our model, we evaluate performance under six different inference-time guidance scales: 1, 2, 3, 3.5, 4, and 5.

As shown in Table. 14 and Fig. 14, guidance scale = 3.5 achieves the best overall performance (FID = 9.27, CLIP = 30.29, HOI = 87.39, DINO = 78.21, SSIM(BG) = 96.57). Correspondingly, the visual results at this setting exhibit the most faithful preservation of the foreground object's appearance. In contrast, lower guidance scales (gs = 1.0 or 2.0) lead to diminished semantic alignment, particularly evident

| Guidance Scale | FID ↓ | CLIP ↑ | HOI ↑ | DINO ↑ | SSIM(BG) ↑ |
|---|---|---|---|---|---|
| gs = 1.0 | 10.11 | 29.42 | 80.01 | 62.33 | 95.25 |
| gs = 2.0 | 9.78 | 29.85 | 81.56 | 71.60 | 95.28 |
| gs = 3.0 | 9.48 | 30.12 | 82.47 | 74.04 | 95.21 |
| **gs = 3.5** | **9.27** | **30.29** | **87.39** | **78.21** | **96.57** |
| gs = 4.0 | 9.39 | 30.19 | 83.91 | 77.56 | 95.89 |
| gs = 5.0 | 9.68 | 29.76 | 81.23 | 76.41 | 96.18 |

Table 14: Performance of our model under different guidance scales during inference. The model is trained with a guidance scale of 1.

Table 11: Ablation on different numbers of views (left) and multi-view generators (right).

| # Views | FID ↓ | CLIP ↑ | HOI ↑ | DINO ↑ | SSIM(BG) ↑ |
|---|---|---|---|---|---|
| 1(No multi-view) | 11.55 | 29.52 | 81.32 | 68.83 | 95.83 |
| 2 | 10.22 | 29.55 | 83.89 | 69.73 | 95.86 |
| 3 | 10.19 | 29.81 | 85.08 | 70.26 | 95.87 |
| 4 | 9.54 | 30.21 | 85.19 | 71.63 | 96.03 |
| 5 | 9.29 | 30.23 | 86.07 | 74.19 | 96.21 |
| **6** | **9.27** | **30.29** | **87.39** | **78.21** | **96.57** |

| Method | FID ↓ | CLIP ↑ | HOI ↑ | DINO ↑ | SSIM(BG) ↑ |
|---|---|---|---|---|---|
| No multi-view | 11.55 | 29.52 | 81.32 | 68.83 | 95.83 |
| **Zero123+**[55] | **9.27** | **30.29** | **87.39** | **78.21** | **96.57** |
| SV3D[73] | 9.89 | 29.85 | 84.98 | 75.26 | 96.01 |
| ViewDiff[21] | 10.20 | 29.99 | 86.19 | 74.63 | 95.98 |

Table 13: Ablation study on different ID encoder backbones

| Backbone | FID ↓ | CLIP ↑ | HOI ↑ | DINO ↑ | CLIP-I ↑ | SSIM(BG) ↑ |
|---|---|---|---|---|---|---|
| VAE [51] | 9.98 | 29.72 | 82.73 | 67.33 | 78.38 | 95.98 |
| CLIP [62] | 9.55 | 30.17 | 85.24 | 75.72 | 87.79 | 96.53 |
| **DINOv2** [43] | **9.27** | **30.29** | **87.39** | **78.21** | **90.25** | **96.57** |

in the foreground regions, as reflected by lower DINO scores. Increasing the scale beyond 3.5 (*e.g.*, gs = 4.0 or 5.0) results in subtle declines in both quantitative scores and foreground object consistency.

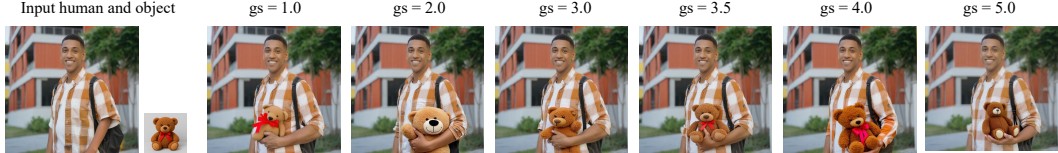

Figure 14: Ablation study on different guidance scales (denoted as gs) during inference.

# I   Comparison with Multi-Modality Models

We compare our method with recent state-of-the-art multi-modality models, including *GPT-4o*[52], *Grok3*[86], and *MidJourney V7* [50]. All models receive identical inputs: a foreground object, a background human image, a designated interaction region, and a corresponding text prompt.

Qualitative results reveal clear limitations in existing models. GPT-4o and MidJourney V7 frequently fail to generate consistent foreground objects (*e.g.*, Row 2(b), Rows 2–3(d) in Fig. 15). Grok3 and MidJourney V7 struggle to preserve the background human and scene details (Rows 1–3(c–d)). In addition, GPT-4o may struggle to accurately model interactions under complex scenarios (see Row 1(b)).

Quantitatively, our method outperforms all baselines across five key metrics. It achieves the lowest FID (9.27), highest CLIP score (30.29), HOI score (87.39), DINO score (78.21) and SSIM(BG) score (96.57). This demonstrate that our method delivers more harmonious human-object interactions and consistent appearances.

# J   Additional Comparison with Image Composition Methods

In addition to the nine methods compared in the main paper, we conducted further comparisons with five additional state-of-the-art image composition methods: DreamFuse [26], InsertAnything [66], MimicBrush [8], Bifrost [36] and DreamRelation [63]. For fairness, all methods with publicly available training code were retrained or fine-tuned on our dataset.

Fig. 17 shows qualitative comparisons. DreamFuse and InsertAnything generate visually faithful foreground objects, but often fail to model realistic human-object interactions (see Rows 2–4 in Fig.17(b–c)). DreamRelation produces interaction-like gestures, yet struggles to preserve the visual consistency of the foreground object and background human (Rows 1–4 in Fig.17(f)). MimicBrush and Bifrost, on the other hand, produce neither convincing interactions nor accurate object appearances (Fig. 17(d–e)). In contrast, our method generates diverse and harmonious interactions while maintaining the consistent appearance of both the foreground and the background.

Table 15: Qualitative comparison with recent state-of-the-art multi-modality models.

| Method | FID↓ | CLIP↑ | HOI↑ | DINO↑ | SSIM(BG)↑ |
|---|---|---|---|---|---|
| Grok3 [86] | 13.27 | 29.07 | 65.03 | 57.02 | 58.25 |
| GPT-4o [52] | 9.98 | 29.35 | 75.22 | 65.23 | 47.22 |
| MidJourney V7 [50] | 10.85 | 29.87 | 73.45 | 60.18 | 41.34 |
| **Ours** | **9.27** | **30.29** | **87.39** | **78.21** | **96.57** |

Figure 15: Quantitative comparison with recent state-of-the-art multi-modality models. The prompts for the above three cases are: "A woman is riding a horse","A girl is holding a stack of books", "A model is presenting a skincare bottle".

Table. 16 provides quantitative results. Our method achieves the best FID (9.27), CLIP-Score (30.29), HOI-Score (87.39), and DINO-Score (78.21), indicating superior image quality, semantic alignment, interaction quality and appearance consistency. User study results further validate our approach, ranking it highest in image quality (IQ), interaction harmonization (IH), and appearance preservation (AP), with all scores significantly outperforming other methods.

# K   Additional Results of *HOComp*

Fig. 17 shows additional qualitative results of our method. Each example includes: (1) Top: the final composited image, (2) Bottom: the input background human and foreground object. These results demonstrate that our method produces natural and plausible human-object interactions while maintaining visual consistency of both the foreground object and the background human.

# L   Ethical Considerations

**Human Subjects and Informed Consent**

Our institution currently does not maintain a formal Institutional Review Board (IRB) or equivalent ethics committee. To ensure compliance with international ethical norms, we conducted an internal review modeled on IRB standards and assessed the study along four key dimensions: privacy, informed consent, participant protection, and data security.

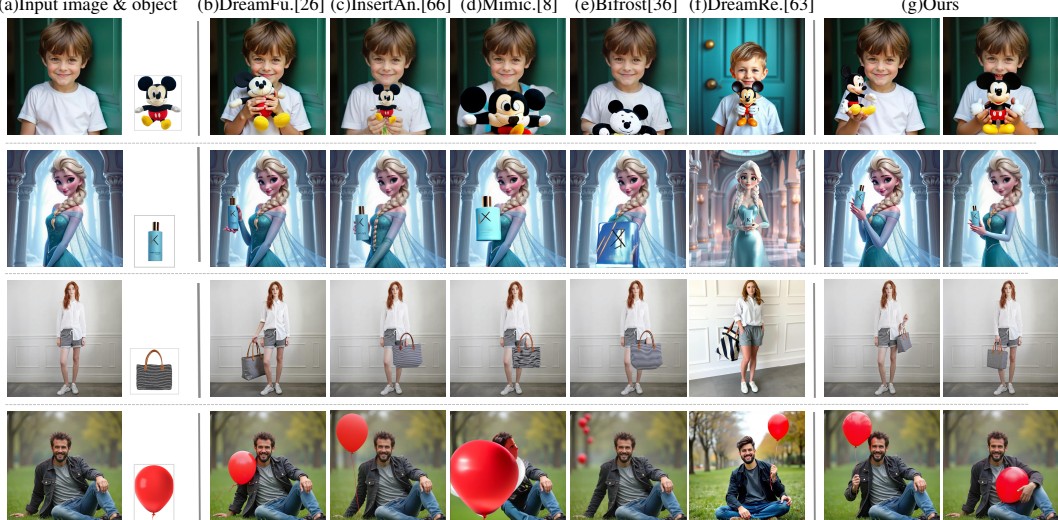

| (a)Input image & object | (b)DreamFu.[26] | (c)InsertAn.[66] | (d)Mimic.[8] | (e)Bifrost[36] | (f)DreamRe.[63] | (g)Ours |
|---|---|---|---|---|---|---|

Figure 16: Additional qualitative comparisons of our ***HOComp*** with 5 SOTA methods. The prompts for the above four examples are: "A boy is holding a mickey mouse toy", "A girl is showing a perfume bottle", "A woman is lifting a bag", and "A sitting man is holding a balloon".

Table 16: Additional quantitative comparison of our method with 5 SOTA methods. The best and second-best results are highlighted in **bold** and underline, respectively. Training or tuning-based methods without released training codes are marked with a [†].

| Category | Metrics | DreamFuse[†] [26] | InsertAnything[†] [66] | MimicBrush[†] [8] | Bifrost[†] [36] | DreamRelation [63] | Ours |
|---|---|---|---|---|---|---|---|
| Automatic | FID ↓ | 13.35 | 10.72 | 15.88 | 16.21 | 15.85 | **9.27** |
| | CLIP-Score ↑ | 29.53 | 29.76 | 28.62 | 28.17 | 28.55 | **30.29** |
| | HOI-Score ↑ | 63.75 | 58.85 | 36.04 | 38.98 | 52.66 | **87.39** |
| | DINO-Score ↑ | 44.89 | 64.52 | 40.67 | 42.02 | 37.07 | **78.21** |
| | SSIM(BG) ↑ | 93.23 | 92.19 | 84.56 | 88.11 | 25.19 | **96.57** |
| User study | IQ ↓ | 3.10 | 2.88 | 4.80 | 5.25 | 3.85 | **1.12** |
| | IH ↓ | 2.28 | 2.43 | 6.00 | 5.95 | 3.27 | **1.07** |
| | AP ↓ | 2.89 | 2.43 | 4.33 | 4.44 | 5.90 | **1.01** |

The user study was conducted via anonymous online questionnaires, in which participants were asked only to rank composite images by visual quality and realism. No identifiable or demographic data were collected. All participants were adult volunteers who provided explicit informed consent before participation and were compensated with an equivalent of US$8 for approximately 15 minutes of participation—exceeding the local minimum wage. Participation was voluntary, and participants could withdraw at any time without penalty. All data were stored on secure, password-protected institutional servers and will be permanently deleted after completion of the analysis.

**Data Licensing and Source Transparency**

Our data usage strictly adheres to the licenses and terms of all sources involved:

- The **HICO-DET**[5] dataset is distributed under the MIT License, permitting unrestricted research and redistribution use.

- The **HOIBench** dataset is constructed using images from Internet, whose licenses allow editing, derivative works, and non-restrictive reuse of images—including those depicting people—without requiring additional permissions.

We employ these resources solely for academic research. To safeguard privacy, any external user wishing to access our dataset must sign an agreement restricting use to non-commercial research and explicitly prohibiting any misuse, defamation, or violation of depicted individuals' rights. We will also publish a clear data-use policy to ensure transparency and traceability of all derivative works.

**Privacy Protection and Consent Risks**

Because the datasets include human figures, we recognize potential privacy and consent issues, especially regarding secondary use of images beyond their original intent. To mitigate such risks, we (1) use all images under licenses that allow derivative use, (2) avoid any manual modification that changes the identity or context of individuals in an offensive or misleading way, and (3) provide clear disclaimers in our documentation that prohibit using HOComp for identity-related, defamatory, or deceptive content.

**Misuse Risks and Responsible Deployment**

We acknowledge that HOComp, like other generative models, carries potential misuse risks such as creating deceptive content or inappropriate composites involving real individuals. To address these risks, we adopt the following safeguards:

- Embedding **invisible watermarks or provenance metadata** in generated images to support detection and accountability.
- Releasing the model and code only for **academic research** under a license that explicitly prohibits deceptive, defamatory, or privacy-violating applications.
- Implementing **content filters** to prevent generation of sensitive or harmful scenarios (e.g., involving weapons, religion, or politics).
- Including **explicit user guidelines and documentation** that highlight responsible use principles and ethical restrictions.

**Environmental Considerations**

Our method fine-tunes existing pre-trained models rather than training from scratch, substantially reducing energy consumption and the carbon footprint associated with large-scale model training. We estimate our total GPU usage to be less than 15% of that required for comparable baseline models trained from scratch.

**Commitment to Responsible AI Research**

We are committed to transparency, accountability, and the responsible use of generative technologies. Our future work will continue to emphasize ethical risk assessment, dataset documentation, and open research practices consistent with the broader goals of trustworthy and socially beneficial AI development.

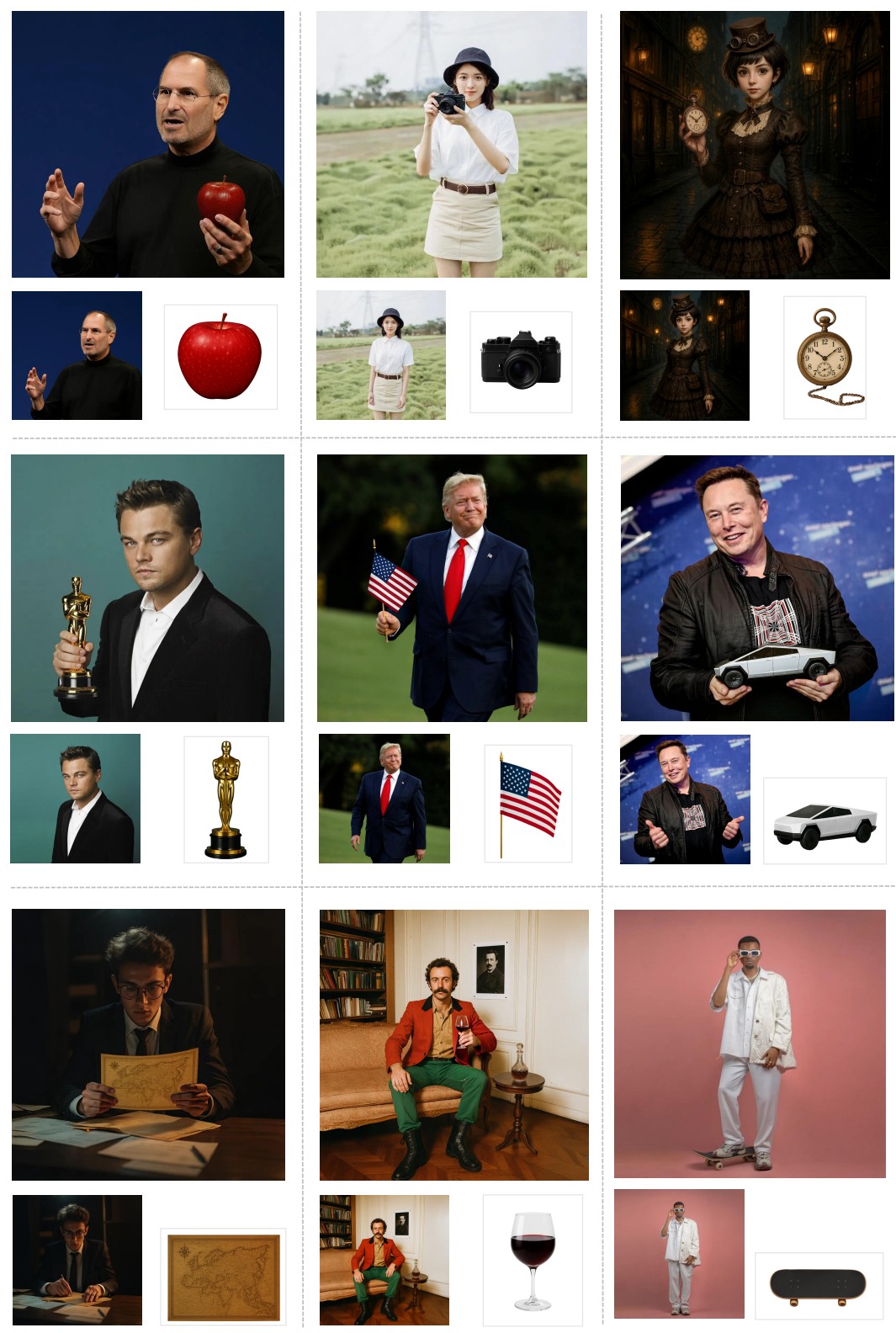

Figure 17: Additional qualitative results of ***HOComp***. Each example includes: (1) Top: the final composited image, (2) Bottom: the input background human and foreground object.

