# OpenReview forum: "HOComp: Interaction-Aware Human-Object Composition"
_NeurIPS.cc/2025/Conference — NeurIPS 2025 poster_

### Official Review · Reviewer_zMtM · 2025-06-29

**Clarity:** 3
**Significance:** 3
**Originality:** 3
**Rating:** 4
**Confidence:** 4

**Summary:**

The paper introduces HOComp, a diffusion-based framework that composites a foreground object into a human-centric background image. A multimodal large language model is used to choose an interaction type (e.g., holding, kicking) and a plausible spatial region on the person. The authors propose a multi-view appearance loss, and a background consistency loss to ensure consistent shapes/textures of the
foreground and faithful reproduction of the background human, and propose Detail-Consistent Appearance Preservation (DCAP) module for preserving appearance. Also, the authors propose a new dataset, IHOC for the human object composition task. Experimental results show competitive results compared with previous image composition methods.

**Questions:**

What is the MLLM's influence on the inference performance? Could you compare using MLLM with other vision-based methods to detect the overlap regions?

**Ethical Concerns:**

["NO or VERY MINOR ethics concerns only"]

**Final Justification:**

The authors partially address my concerns, so I remain my initial positive rating.

**Limitations:**

See weakness

**Quality:**

3

**Strengths And Weaknesses:**

Strengh:
(1) The paper is well-written and easy to follow.
(2) The motivation of human-object interaction in customization task is interesting.

Weakness:
(1) Does this model support multi-subject, multi-person composition? For instance, 2 persons and 2 objects. I am willing to see these results in the rebuttal.
(2) The impact of the area of overlap region to the model's performance is not explored in the paper. What if there is large overlap regions between human and subjects? I suggest the authors do such experiments compared with previous methods in the rebuttal.

---

> ### Author Rebuttal · Authors · 2025-07-31
>
> We thank this reviewer for the positive evaluation and appreciate the recognition in the paper’s clarity and the motivation for the customization of the human-object interaction task. We address each of the questions below. (Abbreviations used: Q/W refer to Question/Weakness.)
>
> ---
>
> ## W1: Support for Multi-object and Multi-person Composition
>
> To evaluate our model’s ability to handle multi-object and multi-person compositions, we have constructed a dedicated test set, since the current HOIBench does not include such cases. Our new benchmark covers two scenarios: (1) two persons with two objects and (2) three persons with three objects.
>
> **Test Set Construction:**
> We first collected 40 images from the internet containing multiple persons: 20 images with two persons, and 20 images with three persons. These images were selected to ensure diversity in human appearance, pose, and clothing.
>
> For **Two Persons – Two Objects:**  To broadly generate interaction-object pairs, we have adopted the 117 interaction types defined in HICO-DET. For each interaction type, we prompt GPT-4o to infer a plausible foreground object (e.g., “playing”→“guitar”). A concise textual description is then used to retrieve a representative object image from the internet, resulting in 117 interaction–object pairs. For the two-person scenario, each person in an image is randomly assigned a unique interaction–object pair from this pool, ensuring a wide variety of combinations. For each of the 20 two-person images, we repeat this assignment process 10 times with different pairs, resulting in 200 two-person, two-object test cases.
>
> For **Three Persons – Three Objects:**  Similarly, for three-person images, each person is assigned a unique interaction–object pair, randomly sampled from the 117 pairs described above. This process is repeated for each three-person image, resulting in 200 three-person, three-object cases.
>
> **Test Results:**
> We have evaluated our model using the new test set of 400 cases (200/scenario) and compared its performance with the top three methods from the main paper. We compute the HOI-Score and DINO-Score for each individual person-object interaction pair, and then take the average of all these scores across all interaction pairs.
>
> As shown in the following table, our approach achieves the best results in both scenarios and on all metrics.
>
>
> | Category               | Methods     | FID↓   | CLIP-Score↑ | HOI-Score↑ | DINO-Score↑ | SSIM(BG)↑ |
> |------------------------|------------|--------|-------------|------------|-------------|-----------|
> | 2 persons & 2 objects  | AnyDoor    | 22.38  | 26.86       | 23.12      | 55.46       | 83.19     |
> |                        | OmniGen.   | 15.98  | 29.03       | 51.45      | 38.66       | 75.91     |
> |                        | GPT-4o     | 11.13  | 29.18       | 73.12      | 63.17       | 25.50     |
> |                        | **Ours**   | **10.25** | **29.87** | **83.61** | **73.48**   | **93.25** |
> | 3 persons & 3 objects  | AnyDoor    | 24.17  | 26.03       | 22.98      | 52.07       | 80.11     |
> |                        | OmniGen.   | 18.25  | 28.14       | 49.01      | 35.85       | 72.14     |
> |                        | GPT-4o     | 13.46  | 28.85       | 70.88      | 55.14       | 19.80     |
> |                        | **Ours**   | **10.43** | **29.76** | **82.98** | **73.09**   | **91.52** |
>
> ---
>
> ## W2: The Impact of the Area of Interaction Region
>
> We evaluated HOComp's performance on cases with exceptionally large interaction regions, such as those illustrated in Fig.2 (Supplementary) (e.g., "a man is playing a guitar," where the interaction covers over 70% of the image). We have identified 82 images in HOIBench where the interaction region occupies more than 70% of the total image area, and compared HOComp with previous methods on this subset. As shown in the following table, HOComp achieves substantially better results than other baselines across all metrics in large interaction region cases.
>
> | Method    | FID↓   | CLIP-Score↑ | HOI-Score↑ | DINO-Score↑ | SSIM(BG)↑ |
> |-----------|--------|-------------|------------|-------------|-----------|
> | AnyDoor   | 21.80  | 26.93       | 22.96      | 44.62       | 85.78     |
> | OmniGen.  | 15.31  | 29.14       | 55.41      | 38.45       | 72.83     |
> | UniCom.   | 13.55  | 29.17       | 54.22      | 49.37       | 84.02     |
> | GPT-4o    | 10.03  | 29.12       | 74.01      | 63.39       | 32.91     |
> | **Ours**  | **9.44** | **30.26** | **85.11** | **77.48**   | **95.43** |
>
> ---
>
> ## Q1 & Q2: MLLMs' Influence on Inference Performance and Comparison with Vision-Based Methods to Detect Interaction Regions
>
> At MPRG's coarse level, it leverages MLLMs to automatically identify suitable interaction types/regions via multi-stage querying. In some rare cases, MLLMs may fail to accurately predict the interaction regions. We provide a detailed analysis of these failure modes and potential solutions below.
>
> In our HOIBench experiments, MLLMs (e.g., GPT-4o) identified reasonable interaction types in **100%** of samples. The corresponding interaction region was predicted correctly in **91.33%** of cases (548/600). For the remaining **8.67%** (52/600), two main failure modes are observed:
> 1. **Mismatch Between Interaction Regions and Types (6% of cases):** When multiple plausible interaction types exist, MLLMs may misassign interaction regions. For example, if the object is sunglasses and the interaction type is "hold", the model may incorrectly assign the region around the eyes (i.e., "wear" action). While the generated image depicts a person wearing sunglasses with harmonious interactions and consistent appearance, the predicted interaction type is inconsistent.
> 2. **Incorrect Interaction Region Size (2.67% of cases):** In these cases, the predicted interaction region does not fully cover the area, including the object and the human body part, for modification. As shown in Fig.6 (main paper), this can hinder generation of correct interactions.
>
> To address these two issues, we explored the following solutions:
> - **Additional Input Conditions:** To alleviate the mismatching issue, we can further incorporate human pose priors (from a pose estimator) as an additional prior to MLLMs. For GPT-4o, the region prediction accuracy increased from **91.33%** to **96.5%**. This improvement can be attributed to the introduction of explicit keypoint information, which provides precise localization of body parts such as the face and hands.
> - **Training with Noisy Data:** To mitigate the impact of incorrect interaction region sizes, we introduce noisy data during training.
>   We first generated 1,000 accurate input data samples. In accordance with the observed error rate, we then moved the bounding boxes of 700 samples to create mismatches with the interaction types (e.g., moving the bounding box of a “soccer ball” with a “kick” action to cover the hand, which may correspond to a “hold” action). Additionally, we reduced the bounding boxes of 300 samples so that they no longer fully covered the interaction object or the required body movement. We fine-tuned the previously pretrained HOComp model for 3,000 steps using these noisy samples alongside the original IHOC dataset. After retraining, we re-evaluated our method using the 52 failure cases. As shown in the following tables, training with noisy data substantially improved the model's performance even when MLLMs mispredicted the interaction region.
>
> | Training Strategy     | FID↓   | CLIP-Score↑ | HOI-Score↑ | DINO-Score↑ | SSIM(BG)↑ |
> |----------------------|--------|-------------|------------|-------------|-----------|
> | Without Noisy Data   | 11.58  | 29.55       | 38.62      | 70.02       | 62.01     |
> | With Noisy Data      | **10.03** | **29.82** | **77.89** | **74.29**   | **88.93** |
>
> Qualitative results and further experimental details will be included in the Supplemental.
>
> **Comparison using MLLM with other vision-based methods to detect interaction regions**: Vision-based methods face significant challenges in this task, as they not only need to localize humans and objects but also infer plausible interaction types and predict specific body parts/regions involved in the interaction.
>
> To the best of our knowledge, existing vision-based approaches [1-3] are limited: they typically operate on images where the interaction already exists (e.g., our composited results), and can only detect existing relationships or involved regions. They generally lack the ability to perform open-ended reasoning to predict plausible interaction types/regions for images where no interaction has yet occurred, given only an isolated person and a reference object.
>
> We have also compared the performance of different MLLMs, as shown in the following table. The accuracy of interaction type/region predictions is computed through a user study involving 20 participants, who evaluate the correctness of MLLM predictions. The correct predictions are counted, and the average accuracy per participant is computed.
>
> | MLLMs        | Interaction Type Prediction Accuracy | Interaction Region Prediction Accuracy |
> |--------------|-------------------------------------|---------------------------------------|
> | GPT-4o       | **100%**                            | **91.33%**                            |
> | Gemini2.5Pro | 99.67%                              | 86.83%                                |
> | Grok4        | 99.5%                               | 88.67%                                |
>
> ---
>
> ## References
>
> [1] Grounding DINO: Marrying DINO with Grounded Pre-Training for Open-Set Object Detection, ECCV'24
> [2] Human-Object Interaction Detection Collaborated with Large Relation-driven Diffusion Models, NeurIPS'24.
> [3] OneRef: Unified One-tower Expression Grounding and Segmentation with Mask Referring Modeling, NeurIPS'24.

---

> > ### Comment · Reviewer_zMtM · 2025-08-09
> >
> > The authors partially address my concerns, so I tend to remain my initial positive rating. I suggest the authors to add these experiments in the final version of paper.

---

> > > ### Author Response · Authors · 2025-08-09
> > > **Thank you for your suggestion and positive evaluation.**
> > >
> > > We sincerely appreciate your continued positive evaluation and constructive feedback.  We will include these additional experiments in the final version of the paper as recommended.

---

### Official Review · Reviewer_7QgL · 2025-06-30

**Clarity:** 3
**Significance:** 3
**Originality:** 3
**Rating:** 4
**Confidence:** 4

**Summary:**

This paper introduces HOComp, a novel framework for human-object image composition that enables natural and realistic interactions between humans and inserted foreground objects. Unlike prior methods that rely on manually defined regions and text prompts, HOComp autonomously determines the interaction type and region using Multimodal Large Language Models (MLLMs) like GPT-4o, and enforces pose and appearance consistency using several novel loss functions and mechanisms.

**Questions:**

1. Several previous works, such as CG-HOI and CHOIS, have annotated the BEHAVE dataset with textual descriptions, and datasets like OMOMO have further extended this line of research into 3D human-object interactions with textual annotations. However, the distinction between the proposed dataset and these prior efforts remains unclear. The authors are encouraged to clarify what specific advantages or novel features their dataset introduces.

2. Is affordance estimation explicitly required in the proposed HOI generation method? The paper would benefit from a clarification on this point.

**Ethical Concerns:**

["Major Concern: Data privacy, copyright, and consent"]

**Final Justification:**

After considering the rebuttal and discussions, I maintain my original evaluation with a few clarifications.

The authors have satisfactorily addressed several concerns.

However, one key issue remains partially unresolved: the lack of direct comparison with recent 3D-based HOI synthesis methods such as CG-HOI and HOIAnimator. To further strengthen the work, I encourage the authors to include such comparisons in future revisions.

**Limitations:**

1. The paper lacks a comprehensive discussion of recent related work. In particular, several relevant works published in 2024 have not been acknowledged or discussed, including CG-HOI, HOIAnimator, InterFusion, Controllable HOI Synthesis, and F-HOI. These recent methods contribute significantly to the field of human-object interaction generation and text-driven synthesis.

 2. To better understand the generalization ability of the proposed model, it may be helpful to evaluate its performance on interactions involving completely unseen objects that are not included in the BEHAVE or OMOMO datasets. If the dataset is skewed toward certain interaction types, there is a risk that the model may overfit to these patterns.

**Quality:**

3

**Strengths And Weaknesses:**

Strengths:

1. A new approach for interaction-aware human-object composition that ensures realistic, coherent, and visually consistent integrations of foreground objects into human-centric images.
2. The first large-scale dataset tailored for this task, including paired before/after images, interaction masks, text prompts, and detailed annotations.

Weaknesses:

1. The authors claim that the model can produce physically plausible results, yet no physical constraints are integrated to guarantee that.
2. There is a lack of comparison of human-object interaction generation methods, making it difficult to evaluate the effect of the proposed method in this domain.
3.  To more accurately assess the physical plausibility of the generated motions, it may be beneficial to incorporate a more sophisticated metric for measuring interpenetration, such as "Intersection Volume." This metric directly quantifies the volume of penetration between interacting bodies and can provide a more precise evaluation of physical correctness. For reference, the paper "Physics-aware Hand-object Interaction Denoising" (CVPR 2024) offers a detailed discussion and implementation of such metrics, which may serve as a useful guideline.

---

> ### Author Rebuttal · Authors · 2025-07-31
>
> We thank this reviewer for the positive review and useful comments. We appreciate your recognition of our IHOC dataset. We address each of the questions below. (Abbreviations used: Q/W/L refer to Question/Weakness/Limitation.)
>
> ---
>
> ## W1: Physical Constraints in Our Model
>
> We would like to clarify that our approach utilizes pose-guided supervision to enforce physical constraints in the generated images. Specifically, during the training of HOComp, we impose human poses constraints within the interaction regions, using pose keypoints as explicit supervision. This helps the model learn physical priors that ensure realistic interactions between the foreground object and the background person.
>
> While it is possible to employ reconstruction methods[2] to generate 3D hand meshes and apply stronger physical constraints on specific body parts (e.g., hands), our model is not limited to hand interactions alone. As shown in Fig.3(c) (Supplemental), samples involving hand interactions alone comprise 54.3% of our training data, with other body parts (e.g., legs) also playing a significant role in the interactions in the remaining samples.
>
> If we were to impose 3D constraints on the entire body, it would have introduced additional modeling errors and substantially increased computational complexity during training. Instead, we use DWPose[1], a robust pose estimator to capture landmarks across the entire body. DWPose predicts multiple hand keypoints and key points for major joints (e.g., each finger contains four keypoints from fingertip to base, and each joint has one keypoint). This approach allows our model to accurately capture human body movements across a variety of interaction types while maintaining computational efficiency.
>
> ---
>
> ## W2&L1: Lack of Discussion and Comparison of Recent Related Works
>
> We thank the reviewer for pointing out relevant works on human-object interaction generation. We would like to clarify that the methods mentioned (CG-HOI, HOIAnimator, InterFusion, Controllable HOI Synthesis, and F-HOI) primarily focus on 3D human-object interaction generation, which aims to generate physically plausible 3D human-object interactions. These methods typically generate 3D models or sequences based on the interaction description, differing significantly from our approach, which focuses on generating 2D composited images.
>
> Our work addresses a distinct task of generating 2D composited images with not only harmonious human-object interactions, but also consistent foreground object and the background person appearances. Existing methods fall short in addressing our challenges. Although human-object interaction image generation methods can generate images with good human-object interactions, which is discussed in Section2 (main paper). One key distinction is that most of the referenced methods do not specify particular reference humans and objects (except for DreamRelation[3] and PersonaHOI[4]). Further, these methods do not emphasize the need to maintain consistent appearances of both humans and objects, in addition to generating harmonious interactions, which is a core aspect of our approach. Because PersonaHOI has not made their official code available for direct comparison, we have included comparisons with DreamRelation in Tab.8 and Fig.10 of the Supplemental.
>
> We will incorporate the above discussions in our revision.
>
> ---
>
> ## W3: Incorporate a More Sophisticated Metric to Assess the Physical Plausibility of the Generated Motions
>
> As suggested, we have thoroughly examined the six metrics, MPJPE and MPVPE (for hand pose/shape reconstruction accuracy), IV and PD (for physical plausibility and penetration degree), Contact IoU (for contact area accuracy), and Plausible Rate (for overall plausibility and manipulation credibility), used in ``Physics-aware Hand-object Interaction Denoising'' (CVPR'24).
>
> However, all of these metrics require the generated 3D mesh for computation. Except for "Intersection Volume'', the other metrics additionally require access to the corresponding GT 3D mesh for evaluation. Our method produces only 2D composited images as output and does not include corresponding 3D GT during inference. While it is theoretically possible to reconstruct 3D meshes from 2D images and then compute Intersection Volume without the need for 3D GT, such reconstruction can introduce significant errors, especially given the diverse interaction types and occlusions present in our task.
> Nevertheless, we will explore if there are more appropriate metrics for such an evaluation.
>
> ---
>
> ## Q1: Difference from Prior Human-object Interaction Datasets
>
> The key distinctions and advantages of our dataset compared to BEHAVE and OMOMO are as follows:
>
> **Task and Modality:** Our dataset is the first large-scale resource specifically tailored for the interaction-aware human-object composition task, providing paired pre-/post-interaction image data with comprehensive annotations. In contrast, BEHAVE and OMOMO focus primarily on 3D human-object interaction generation, providing 3D meshes of humans and objects. While BEHAVE contains some RGB images during interaction, it lacks the crucial pre-interaction data required for modeling before/after composition.
>
> **Scale and Diversity:** Our dataset covers a significantly broader range of human-object interactions, actions, and scenes. Compared to BEHAVE (8 subjects, 20 objects) and OMOMO (17 subjects, 15 objects), our dataset includes 11,700 unique samples, each with a different human subject. We provide 117 types of interactions and 342 distinct foreground object categories, encompassing diverse human poses, viewpoints, both indoor and outdoor environments, and a wide variety of object categories, sizes, and interaction body parts. In addition, our dataset features a broad spectrum of image styles. Comprehensive statistics and visualizations can be found in Fig.3 (Supplemental).
>
> In summary, our dataset uniquely enables research on image-based, interaction-aware human-object composition with a level of scale, diversity, and paired pre/post-interaction coverage not available in prior datasets. We will add this discussion in our revision.
>
> ---
>
> ## Q2: "Is affordance estimation explicitly required in the proposed method?"
>
> Our method does not explicitly involve affordance estimation. Instead, we rely on the MLLM to infer the appropriate human-object interaction based on the visual input of a background human image and a foreground object image. The MLLM performs reasoning on these inputs to predict the interaction type, without the need for explicit affordance estimation. In our HOIBench evaluation, MLLMs (e.g., GPT-4o) correctly predicted plausible interaction types in 100% of the tested cases.
>
> ---
>
> ## L2: Generalization to Interactions with Out-of-Distribution Objects
>
> Our method performs well on objects not included in the BEHAVE or OMOMO datasets. For example, all examples shown in Fig.1 in the main paper, as well as those in Fig.4 (rows 1, 2, and 4), involve object categories that are absent from both BEHAVE and OMOMO. Nevertheless, our method is able to generate satisfactory results on these cases.
>
> We would like to clarify that, when constructing our dataset, we did **not** select object types based on those present in BEHAVE or OMOMO. Specifically, BEHAVE contains only 20 objects and OMOMO contains 15 objects. In contrast, the object types in our training set were determined using an MLLM, which inferred potential object categories according to 117 interaction types (such as ''hold'', ''play''). These pairings (e.g., ''hold-bag'', ''play-guitar'') were then manually verified for plausibility, resulting in a total of **342 distinct object categories** in our dataset.
>
> In our **HOIBench** evaluation, we cover all 117 interaction types and 108 object categories. While some categories may overlap with those in the training set, we ensured that **both the human subjects and the object images in the test set are completely disjoint from those in the training set**. We will provide more visual results in the supplementary materials.
>
> ---
>
> ## Ethical Concerns on Data Privacy, Copyright, and Consent
>
> All images in our training set are sourced from the public HICO-DET dataset, images generated by the T2I model FLUX.1 [dev], or openly licensed free image websites such as Unsplash and Pexels. The HOIBench also exclusively uses images from freely available public websites. Therefore, there are no ethical issues for our dataset. We will release our dataset upon acceptance of the paper.
>
>
> ---
>
>
> ## References
>
> [1] Effective Whole-body Pose Estimation with Two-stages Distillation, ICCV'23.
> [2] Reconstructing Hands in 3D with Transformers, CVPR'24.
> [3] Dreamrelation: Bridging customization and relation generation, CVPR'25.
> [4] PersonaHOI: Effortlessly Improving Personalized Face with Human-Object Interaction Generation, CVPR'25.

---

> > ### Comment · Reviewer_7QgL · 2025-08-06
> > **Thank you for the detailed and thoughtful response.**
> >
> > Thank you for your thoughtful response, which addresses most of our concerns. However, there are still some points that remain unclear or could benefit from further clarification:
> >
> > 1. Unclear Physical Constraints: While "pose-guided supervision" is mentioned as a physical constraint, the explanation lacks specific details on how it ensures realistic interactions. The reviewer might expect a more detailed explanation, such as how these constraints prevent object penetration.
> >
> > 2. Computational Trade-Off: While avoiding full-body 3D constraints to reduce complexity is understandable, the response does not fully address whether this compromises physical accuracy, and no empirical data is provided to support this decision.
> >
> > 3. Unclear Contributions: The response does not thoroughly explain why methods like CG-HOI, HOIAnimator, etc., are not directly applicable to the paper’s approach. While it mentions their focus on 3D generation, it does not clarify why their techniques wouldn't provide valuable insights for 2D composited images. Additionally, it's worth noting that while 3D generation focuses on models or sequences, these can be rendered into 2D images. Similarly, sequence-based methods generate frame-by-frame compositions, which aligns with the frame-by-frame nature of 2D compositing in our approach.

---

> > > ### Author Response · Authors · 2025-08-07
> > > **Thank you for your valuable comments and for raising these unclear questions.**
> > >
> > > ## 1. Unclear Physical Constraints:
> > >
> > > To clarify, in our method, pose-guided supervision serves as the primary physical constraint, ensuring realistic human-object interactions by directly penalizing implausible joint positions and discouraging penetration. Additionally, our Shape-aware Attention Modulation(SAAM) further assists by correcting occlusion errors, thereby reducing penetration.
> > >
> > > ### 1.1 Ensuring Correct Joint Positions:
> > >
> > > Pose-guided supervision ensures that predicted keypoints (e.g., joints of fingers or limbs) align with ground-truth keypoints. By optimizing the pose-guided loss, it penalizes physically implausible joint configurations, such as unnatural bending or hyperextension. As a result, the model is explicitly guided during training to learn physically plausible joint-object relationships, ultimately leading to more natural and accurate interactions.
> > >
> > > ### 1.2 Prevention of Object Penetration
> > >
> > > For a clear illustration of the problem, please refer to the example of hand penetrating an object shown in Fig 5, row 2 of paper "Physics-aware Hand-object Interaction Denoising". Object penetration may leads to two observable artifacts:
> > > (i) incorrect occlusion (e.g., object pixels occlude body parts they should not), and
> > > (ii) misaligned keypoints.
> > >
> > > For (i), penetration may leads to incorrect occlusion, where the object erroneously covers parts of the hand (e.g., part of a finger is occluded by the object due to penetration). SAAM (Section 3.3) leverages a foreground object mask to correct these occlusion relationships, ensuring the hand and object are properly layered. This process effectively reduces penetration artifacts at the attention level.
> > >
> > > For (ii), penetration (e.g., a finger passing through a can) results in predicted keypoints deviating from the ground-truth, which always represents a physically valid, non-penetrating pose. This discrepancy is penalized by the pose-guided supervision, further discouraging penetration during training.
> > >
> > > ---
> > >
> > > ## 2. Computational Trade-Off
> > >
> > > Since obtaining 3D ground-truth for the test cases is challenging, we addressed concerns about potential compromises in physical accuracy (due to omitting full-body 3D constraints) via a user study. Specifically, 30 participants evaluated 50 randomly selected cases from each method, judging whether the interactions were physically plausible (i.e., free from penetration, missing contact, or implausible joint bending).
> > >
> > > The proportion of physically plausible cases (“Plausible Rate”) for each method is shown below:
> > >
> > > | Method           | AnyDoor | PbE   | FreeComp. | FreeCustom | PrimeComp. | OmniGen. | GenArt. | UniCom. | GPT-4o | Ours    |
> > > |------------------|---------|-------|-----------|------------|------------|----------|---------|---------|--------|---------|
> > > | Plausible Rate ↑ |  37.6   | 42.5  | 31.5      | 28.8       | 25.1       | 56.5     | 48.7    | 71.6    | 82.4   | **98.9** |
> > >
> > > Our method achieves the highest plausible rate, showing it produces physically plausible interactions without full-body 3D constraints and extra computational cost.
> > >
> > > We recognize the potential benefits of incorporating full-body 3D constraints, and we plan to explore integrating such constraints in future work to further improve physical accuracy.

---

> > > ### Author Response · Authors · 2025-08-07
> > >
> > > ## 3. Unclear Contributions
> > >
> > > While 3D methods such as CG-HOI and HOIAnimator offer important insights for composing realistic interactions—such as leveraging explicit contact modeling, and coordinating human-object motions through learned interaction fields—these approaches fundamentally rely on access to detailed 3D geometry.
> > >
> > >
> > > ### 3.1 Key Differences
> > >
> > > - **Broader Applicability:**
> > >   3D methods require detailed geometry and texture for both humans and objects, which enables high-fidelity interaction rendering. However, acquiring such 3D assets is costly and may result in limited dataset diversity—e.g., OMOMO and BEHAVE datasets contain fewer than 20 object or human categories. In contrast, our approach is fully 2D and does not depend on any mesh-based 3D pipelines, greatly simplifying data acquisition. This enables support for a significantly broader range of interaction types (117) and object categories (342) using only image data.
> > >
> > > - **Distinct Technical Challenges:**
> > > Maintaining consistent appearance for both the human and the object—particularly the changes in object viewpoint and texture caused by pose variations—is considerably more challenging in 2D. While 3D approaches leverage geometric information to address these issues, 2D methods must infer or synthesize such changes without access to explicit geometry.
> > >
> > > ### 3.2 Our Contribution
> > >
> > > Our approach achieves both **consistent appearance** and **hamonious interactions** solely from 2D data, by leveraging learned priors and weak supervision—without requiring any explicit 3D geometry.
> > >
> > > To specifically address the challenge of appearance consistency in 2D compositing, we introduce the **DCAP** module (see Section 3.3). DCAP integrates shape-aware attention modulation, multi-view appearance loss, and background consistency loss, enabling the generation of composited images with consistent object details like shapes and textures, and human appearance, despite the absence of 3D geometry.
> > >
> > > ### 3.3 Limitations of Direct Application
> > >
> > > Although 3D methods can produce 2D projections or sequences, their reliance on high-quality 3D assets severely limits their utility for real-world 2D compositing tasks. In practical scenarios such as advertising or photo editing, obtaining a 3D scan of a specific person or object is often infeasible, whereas collecting photographic references is straightforward. Our method is designed to operate directly on 2D images, producing robust and physically plausible human-object compositions with much broader accessibility and applicability.

---

> > > > ### Comment · Reviewer_7QgL · 2025-08-08
> > > > **Thank you for your response and clarification.**
> > > >
> > > > If possible, including additional comparative results with 3D-based methods such as CG-HOI and HOIAnimator would further strengthen the argument and provide a more comprehensive demonstration of the proposed method's advantages.

---

> > > > > ### Author Response · Authors · 2025-08-08
> > > > > **Thank you for the helpful suggestion**
> > > > >
> > > > > We fully agree that adding comparisons with recent 3D-based methods would strengthen the paper. Unfortunately, CG-HOI, HOIAnimator, and F-HOI do not provide released code, so we are currently attempting comparisons with other 3D-based methods such as InterFusion and CHOIS. These methods, however, differ materially from ours in their input assumptions—InterFusion uses text-only inputs (no reference object), while CHOIS takes mesh-only inputs without texture—making a strictly fair, like-for-like evaluation nontrivial.
> > > > >
> > > > > We will make every effort to complete these experiments and include the results before the discussion deadline. If this is not feasible, we will add the comparative experiments in the revised version of the paper.

---

> > > > > ### Author Response · Authors · 2025-08-09
> > > > >
> > > > > Following your advice, we additionally compare our method with **InterFusion** [1] mentioned in your review.
> > > > >
> > > > > Since InterFusion generates a 3D human-object interaction from text, for a fair comparison, we randomly select 100 text descriptions from HOIBench and generate InterFusion results, which are then rendered into 2D images (denoted as *composited images*). For our method, we follow the same training set construction pipeline described in Section 3.4 of the main paper to obtain the *foreground object image* and *background human image*, and use them as input to generate our results.
> > > > >
> > > > > The quantitative comparison is shown in the table below (DINO-Score and SSIM(BG) are omitted here since InterFusion does not take a reference foreground object or background human as input):
> > > > >
> > > > > | Method       | FID↓  | CLIP-Score↑ | HOI-Score↑ |
> > > > > |--------------|-------|-------------|------------|
> > > > > | InterFusion  | 13.58 | 28.63       | 69.09      |
> > > > > | Ours         | **10.65** | **29.79**       | **74.15**      |
> > > > >
> > > > > As shown, our method achieves superior performance across all metrics, demonstrating clear advantages in generating realistic human-object interaction images.
> > > > >
> > > > > Due to time constraints, we will include comparisons with additional 3D-based methods in the revised version.
> > > > >
> > > > > [1] InterFusion: Text-Driven Generation of 3D Human-Object Interaction, ECCV 2024

---

### Official Review · Reviewer_2Em8 · 2025-07-01

**Clarity:** 3
**Significance:** 3
**Originality:** 3
**Rating:** 4
**Confidence:** 3

**Summary:**

1. HOComp present to integrate a foreground object onto a human-centric background image seamlessly while ensuring harmonious interactions and preserving the visual consistency of both the foreground object and the background person.

2. MRPG utilizes MLLMs to identify the interaction region and interaction type to provide coarse-to-fine constraints, while incorporating human pose landmarks to track action variations and enforcing fine-grained pose constraints. DCAP unifies a shape-aware attention modulation mechanism, a multi-view appearance loss, and a background consistency loss to ensure consistency.

3. This paper introduces a Interaction-aware Human-Object Composition (IHOC) dataset to facilitate HOC task

**Questions:**

The entire framework includes multiple complex modules to ensure the synthesis of seamless interaction-aware compositions. Does the author team plan to open-source the project and the proposed dataset to promote community development?

**Ethical Concerns:**

["NO or VERY MINOR ethics concerns only"]

**Final Justification:**

I would like to express my gratitude to the author for their rebuttal, which effectively addressed the concerns I had raised.

**Limitations:**

yes.

**Paper Formatting Concerns:**

There are no formatting issues.

**Quality:**

3

**Strengths And Weaknesses:**

Strengths

This paper proposes an effective image editing model for compositing a foreground object onto a human-centric background image. It includes two modules: MLLMs-driven Region-based Pose Guidance to identify the interaction and Detail-Consistent Appearance Preservation to ensure consistency. The paper is well-organized and easy to follow. They also reports sufficient qualitative and quantitative results to support their claims.

Weaknesses

The entire process is quite complex, not only involving multiple processing units to encode the features of the foreground and background, but also employing several loss functions to enforce consistency. I am not very sure about the generalizability of HOComp and the reproducibility of the overall architecture.

---

> ### Author Rebuttal · Authors · 2025-07-31
>
> We thank this reviewer for the positive review and constructive comments. We appreciate the recognition of paper organization and clarity, as well as the sufficiency of our qualitative/quantitative results. We address each of the questions below. (Abbreviations used: Q/W refer to Question/Weakness.)
>
> ---
>
> ### Q&W: Concerns about the Complexity of the HOComp Framework.
>
> Although our framework contains multiple components, each component of HOComp fulfills a distinct, essential function: the framework builds on latest diffusion transformers, integrates MLLMs to enforce coarse-level interaction constraints and reduce user input, incorporates a pose estimator during training to optimize human pose generation for harmonious interactions, and employs a multi-view generator to ensure appearance consistency and semantic alignment across viewpoints.
>
> ---
>
> ### Q&W: Generalizability of HOComp.
>
> To address the generalizability concern of our model, we have evaluated HOComp under challenging and diverse conditions.
>
> We first validated our method's performance under three extreme or special cases:
> (1) multi-subject and multi-person compositions,
> (2) extremely large interaction regions, and
> (3) interactions involving completely occluded target objects.
>
> ---
>
> #### 1. Multi-subject and multi-person compositions:
>
> To evaluate our model’s ability to handle multi-object and multi-person compositions, we have constructed a dedicated test set, since the current HOIBench does not include such cases. Our new benchmark covers two scenarios: (1) two persons with two objects and (2) three persons with three objects.
>
> **Test Set Construction:**
> We have collected 40 diverse images (20 with two persons, 20 with three persons) from the internet, and generated 117 interaction–foreground image pairs following the same procedure of constructing HOIBench described in Section 4 (main paper).
>
> For **Two persons, two objects:** Each person in an image is randomly assigned a unique interaction–object pair, ensuring a wide variety of combinations. For each of the 20 two-person images, we repeat this assignment process 10 times with different pairs, resulting in 200 two-person, two-object test cases.
>
> For **Three persons, three objects:** Similarly, each person is assigned a unique interaction–object pair, randomly sampled from the 117 pairs described above. This process is repeated for each three-person image, resulting in 200 three-person, three-object cases.
>
> We have evaluated our model using the new test set of 400 cases (200/scenario) and compared its performance with the top three methods from the main paper. We compute the HOI-Score and DINO-Score for each individual person-object interaction pair, and then take the average of all these scores across all interaction pairs. As shown in the following table, our approach achieves the best results in both scenarios and on all metrics.
>
>
> | Category               | Method     | FID↓   | CLIP-Score↑ | HOI-Score↑ | DINO-Score↑ | SSIM(BG)↑ |
> |------------------------|-----------|--------|-------------|------------|-------------|-----------|
> | 2 persons & 2 objects  | AnyDoor   | 22.38  | 26.86       | 23.12      | 55.46       | 83.19     |
> |                        | OmniGen.  | 15.98  | 29.03       | 51.45      | 38.66       | 75.91     |
> |                        | GPT-4o    | 11.13  | 29.18       | 73.12      | 63.17       | 25.50     |
> |                        | **Ours**  | **10.25** | **29.87** | **83.61**  | **73.48**   | **93.25** |
> | 3 persons & 3 objects  | AnyDoor   | 24.17  | 26.03       | 22.98      | 52.07       | 80.11     |
> |                        | OmniGen.  | 18.25  | 28.14       | 49.01      | 35.85       | 72.14     |
> |                        | GPT-4o    | 13.46  | 28.85       | 70.88      | 55.14       | 19.80     |
> |                        | **Ours**  | **10.43** | **29.76** | **82.98**  | **73.09**   | **91.52** |
>
> ---
>
> #### 2. Extremely large interaction regions:
>
> We have also evaluated HOComp's performance on cases with exceptionally large interaction regions, such as those illustrated in Fig.2 (Supplementary) (e.g., "a man is playing a guitar," where the interaction covers over 70% of the image). We have identified 82 images in HOIBench where the interaction region occupies more than 70% of the total image area, and compared HOComp with previous methods on this subset. As shown in the following table, HOComp achieves substantially better results than other baselines across all metrics in large interaction region cases.
>
>
> | Method    | FID↓   | CLIP-Score↑ | HOI-Score↑ | DINO-Score↑ | SSIM(BG)↑ |
> |-----------|--------|-------------|------------|-------------|-----------|
> | AnyDoor   | 21.80  | 26.93       | 22.96      | 44.62       | 85.78     |
> | OmniGen.  | 15.31  | 29.14       | 55.41      | 38.45       | 72.83     |
> | UniCom.   | 13.55  | 29.17       | 54.22      | 49.37       | 84.02     |
> | GPT-4o    | 10.03  | 29.12       | 74.01      | 63.39       | 32.91     |
> | **Ours**  | **9.44** | **30.26**  | **85.11**  | **77.48**   | **95.43** |
>
> ---
>
> #### 3. Interactions involving completely occluded target objects:
>
> Some actions may cause the interacting object to become entirely occluded in the resulting image, e.g., food being **eaten** (fully inside the mouth) or an object **blocked** by a hand. To evaluate the model’s generalization in such cases, we have curated 50 test instances where the target object is fully occluded after the interaction. We append “the object is occluded because of the human action” to the MLLM-generated descriptions, forming a dedicated evaluation subset.
>
> Standard metrics requiring object visibility (e.g., DINO-Score, HOI-Score) are inapplicable here. Instead, we manually assess the plausibility of the depicted interactions (“Occlusion Accuracy”), focusing on whether the generated images realistically reflect the intended occlusion. The results in the table below clearly demonstrate the superiority of our method over existing works.
>
>
> | Method    | FID↓   | CLIP-Score↑ | Occlusion Accuracy↑ | SSIM(BG)↑ |
> |-----------|--------|-------------|---------------------|-----------|
> | AnyDoor   | 22.95  | 26.85       | 0.08                | 83.52     |
> | OmniGen.  | 15.83  | 29.07       | 0.42                | 71.89     |
> | UniCom.   | 15.58  | 29.06       | 0.28                | 83.62     |
> | GPT-4o    | 10.52  | 29.31       | 0.84                | 25.28     |
> | **Ours**  | **9.82** | **29.87**  | **0.86**            | **94.35** |
>
> We will include these qualitative results in our paper.
>
> ---
>
> **Extension to the User Study:**
> To further demonstrate the generalization of our method to diverse inputs, we have expanded the user study in the main paper from 10 to 50 input cases, covering a broader range of human poses, viewpoints, and object categories. 45 participants have completed the evaluation. As shown below, our method consistently outperforms all baselines across all metrics, indicating strong user preferences for our approach.
>
>
> | Metrics        | AnyDoor | PbE   | FreeComp. | FreeCustom | PrimeComp. | OmniGen. | GenArt. | UniCom. | GPT-4o | Ours   |
> |---------------|---------|-------|-----------|------------|------------|----------|---------|---------|--------|--------|
> | IQ ↓          | 8.28    | 7.64  | 9.66      | 9.16       | 3.14       | _2.49_   | 6.03    | 4.07    | 3.29   | **1.24** |
> | IH ↓          | 8.65    | 8.90  | 8.35      | 5.85       | 6.15       | 5.01     | 5.34    | 2.91    | _2.69_ | **1.15** |
> | AP ↓          | _3.05_  | 5.04  | 7.20      | 6.92       | 6.25       | 4.93     | 6.65    | 8.05    | 5.74   | **1.17** |
>
> ---
>
> Our framework can also be adapted to different foundation models.
> In our original experiments, we used FLUX.1[dev] as the base DiT model. We additionally evaluated HOComp with FLUX.1 Kontext[dev] [1] as the base model. As shown below, HOComp attains superior performance on FLUX.1 Kontext[dev], further demonstrating the adaptability of our approach across base model variants.
>
>
> | Base DiT Models      | FID↓   | CLIP-Score↑ | HOI-Score↑ | DINO-Score↑ | SSIM (BG)↑ |
> |----------------------|--------|-------------|------------|-------------|------------|
> | FLUX.1[dev]          | 9.27   | 30.29       | 87.39      | 78.21       | 96.57      |
> | FLUX.1 Kontext[dev]  | **9.19** | **30.32**  | **89.21**  | **81.13**   | **97.34**  |
>
> ---
>
> ### Q&W: Reproducibility of our HOComp.
>
> We will release all codes, the IHOC dataset, and HOIBench upon acceptance, to facilitate reproducibility.
>
> ---
>
> ### References
>
> [1] FLUX.1 Kontext: Flow Matching for In-Context Image Generation and Editing in Latent Space, arXiv 2506.15742.

---

### Official Review · Reviewer_gQQw · 2025-07-03

**Clarity:** 3
**Significance:** 2
**Originality:** 2
**Rating:** 4
**Confidence:** 3

**Summary:**

This paper introduces HOComp for interaction-aware human-object composition. Existing methods often fail to generate plausible human poses or preserve appearance. HOComp addresses this with two key components. First, MLLM-driven Region-based Pose Guidance (MRPG) automatically determines the interaction type and region to guide pose generation. Second, Detail-Consistent Appearance Preservation (DCAP) uses multiple losses and attention modulation to maintain foreground and background consistency. The authors also introduce the IHOC dataset for this task. Experiments show their method significantly outperforms state-of-the-art approaches in generating harmonious and realistic compositions.

**Questions:**

1. Regarding the MRPG module's reliance on GPT-4o: You are commendably transparent about the 91.33% success rate. However, a ~9% failure rate at the first step is a significant practical limitation. Could you provide a more detailed analysis of these failure modes? For example, what are the common types of errors (e.g., incorrect body part, illogical interaction type), and how does the rest of the pipeline behave when it receives this incorrect guidance?

2. Regarding the dataset construction: The method of creating paired data by using an inpainting model to generate the "before interaction" image raises a critical question about potential model bias. The model might be learning to invert the specific process of the inpainting tool rather than a general interaction generation skill.

**Ethical Concerns:**

["NO or VERY MINOR ethics concerns only"]

**Final Justification:**

I appreciate the author's rebuttal which addressed some of my concerns, so I maintain my original score.

**Limitations:**

Yes

**Quality:**

3

**Strengths And Weaknesses:**

- Strengths

1. The proposed MRPG module is a novel use of MLLMs to automate the generation of high-level interaction semantics (type and region), simplifying the user input process.

2. The creation of the IHOC dataset and the HOIBench benchmark is a significant contribution to the community, providing valuable resources for future research in this area.

3. The quantitative and qualitative results, including a user study, are extensive and demonstrate a clear and substantial improvement over numerous state-of-the-art methods.

- Weaknesses

1. The framework's performance heavily relies on an MLLM for initial guidance. While the authors report a high success rate, the system is still susceptible to failures from this non-differentiable module, which can lead to incorrect or nonsensical interactions.

2. The system architecture is highly complex, integrating multiple models (DiT, MLLM, pose estimator, multi-view generator) and a carefully balanced set of loss functions. This complexity may pose challenges for reproducibility, training stability, and tuning.

3. The IHOC training dataset is partially constructed using other generative models for inpainting backgrounds and completing occluded objects. This introduces a potential data bias, where the model may be implicitly learning to reverse the artifacts of the data-generation models rather than learning a more general composition skill.

---

> ### Author Rebuttal · Authors · 2025-07-31
>
> We thank this reviewer for the positive review and insightful questions. We appreciate the recognition of the novelty of our MRPG module, the value of the IHOC dataset and HOIBench benchmark, and the appreciation of our experimental results. We address each of the questions below. (Abbreviations used: Q/W refer to Question/Weakness.)
>
> ---
>
> ## Q1&W1: Concerns Regarding MRPG's Reliance on MLLMs' Predictions
>
> To address this concern, we provide a detailed analysis of the failure modes and potential solutions.
>
> In our HOIBench experiments, MLLMs (e.g., GPT-4o) identified reasonable interaction types in **100%** of samples. The corresponding interaction region was predicted correctly in **91.33%** of cases (548/600). For the remaining **8.67%** (52/600), two main failure modes are observed:
>
> 1. **Mismatch Between Interaction Regions and Types (6% of cases):**
>    When multiple plausible interaction types exist, MLLMs may misassign interaction regions. For example, if the object is sunglasses and the interaction type is "hold", the model may incorrectly assign the region around the eyes (i.e., "wear" action). While the generated image depicts a person wearing sunglasses with harmonious interactions and consistent appearance, the predicted interaction type is inconsistent.
>
> 2. **Incorrect Interaction Region Size (2.67% of cases):**
>    In these cases, the predicted interaction region does not fully cover the area, including the object and the human body part, for modification. As shown in Fig.6 (main paper), this can hinder generation of correct interactions.
>
> We have explored the following potential solutions to mitigate these two issues:
>
> - **Additional Input Conditions:**
> To alleviate the mismatching issue, we can further incorporate human pose priors (from a pose estimator) as an additional prior to MLLMs.
>
>   For GPT-4o, the region prediction accuracy increased from **91.33%** to **96.5%**. This improvement can be attributed to the introduction of explicit keypoint information, which provides precise localization of body parts such as the face and hands.
>
> - **Training with Noisy Data:**
>   To mitigate the impact of incorrect interaction region sizes, we introduce noisy data during training.
>
>   We first generated 1,000 accurate input data samples. In accordance with the observed error rate, we then moved the bounding boxes of 700 samples to create mismatches with the interaction types (e.g., moving the bounding box of a "soccer ball" with a "kick" action to cover the hand, which may correspond to a "hold" action). Additionally, we reduced the bounding boxes of 300 samples so that they no longer fully covered the interaction object or the required body movement. We fine-tuned the previously pretrained HOComp model for 3,000 steps using these noisy samples alongside the original IHOC dataset. After retraining, we re-evaluated our method using the 52 failure cases. As shown in the following table, training with noisy data substantially improved the model's performance even when MLLMs mispredicted the interaction region.
>
> | **Training Strategy**  | FID↓  | CLIP-Score↑ | HOI-Score↑ | DINO-Score↑ | SSIM (BG)↑ |
> |-----------------------|-------|-------------|------------|-------------|------------|
> | Without Noisy Data    | 11.58 | 29.55       | 38.62      | 70.02       | 62.01      |
> | With Noisy Data       | **10.03** | **29.82**   | **77.89**  | **74.29**   | **88.93**  |
>
> Qualitative results and further experimental details will be included in the Supplemental.
>
> ---
>
> ## W2: Concerns on Framework Complexity and Reproducibility
>
> Although our framework comprises multiple components, each component of HOComp fulfills a distinct, essential function: Our framework builds on state-of-the-art diffusion transformers, introduces MLLMs for coarse-level interaction constraints, and reduces user input requirements. The pose estimator is incorporated during training to optimize the generated human poses, enabling more harmonious interactions. The multi-view generator ensures appearance consistency and semantic alignment across viewpoints.
>
> We will release all codes, the IHOC dataset, and HOIBench upon acceptance, to facilitate reproducibility.
>
> ---
>
> ## Q2&W3: Potential Data Bias in the IHOC Dataset
>
> We understand and appreciate the reviewer’s concern about potential bias introduced by the use of inpainting models to generate pre-interaction images in IHOC dataset.
>
> First, our IHOC dataset is a mixture of real-world data collected from the internet and AIGC-generated content. During training, inpainting is employed to generate pre-interaction images, similar to previous methods [1,2]. In contrast, the ground-truth post-interaction images depict natural human-object interactions sourced from both real-world images and AIGC-generated interaction images.
>
> Second, the vast majority of input images used in our evaluation are real-world photographs, not inpainted data (e.g., rows 1–3 in Fig.1 (main paper), rows 2–3 in Fig.4 (main paper), and rows 1–3 in Fig.9 (supplementary)). The strong performance of our model on these real images demonstrates that it does not rely on or reproduce inpainting-specific biases.
>
> Finally, to further alleviate this concern, we will provide additional results on more real-world test cases, and in future work, we plan to expand our dataset by extracting pre- and post-interaction pairs directly from video footage to ensure even greater realism and diversity.
>
> ---
>
> ### References
>
> 1. Learning Object Placement by Inpainting for Compositional Data Augmentation. ECCV'20.
> 2. ObjectMate: A Recurrence Prior for Object Insertion and Subject‑Driven Generation, ICCV'25.

---

> > ### Comment · Reviewer_gQQw · 2025-08-06
> >
> > I appreciate the author's rebuttal which addressed some of my concerns, so I maintain my original score.

---

> > > ### Author Response · Authors · 2025-08-08
> > > **Thank you for maintaining your positive evaluation**
> > >
> > > Thank you for maintaining your positive evaluation. We truly appreciate your thoughtful feedback and are glad our rebuttal addressed some of your concerns. If you have any further questions or suggestions, we would be more than happy to address them.

---

### Official Review · Reviewer_dYr4 · 2025-07-04

**Clarity:** 3
**Significance:** 3
**Originality:** 3
**Rating:** 4
**Confidence:** 3

**Summary:**

This paper present HOComp, a framework for interaction-aware human-object composition. It leverages MLLM driven region-based pose guidance (MRPG) for constrained human-object interaction, and detail-consistent appearance preservation (DCAP) for maintaining appearance consistency. This paper also introduced the Interaction-aware Human-Object Composition (IHOC) dataset. Extensive experiments demonstrate that HOComp outperforms existing methods in quantitative, qualitative, and subjective evaluations.

**Questions:**

1.Ordinary and unremarkable writing
2.The novelty level is not too high

**Ethical Concerns:**

["NO or VERY MINOR ethics concerns only"]

**Final Justification:**

I appreciate the author's rebuttal which addressed some of my concerns, so I raise my original score to 4 from 3.

**Limitations:**

1.hope have more bad cases analysis

**Quality:**

3

**Strengths And Weaknesses:**

Strengths:
1. This paper propose a new approach for interaction-aware human-object composition, named HOComp, which focuses on seamlessly integrating a foreground object onto a human-centric  background image while ensuring harmonious interactions and preserving the visual consistency of both the foreground object and the background person.
2. HOComp incorporates two innovative designs: MLLMs-driven region-based pose guidance (MRPG) for constraining human-object interaction via a coarse-to-fine strategy,  and detail-consistent appearance preservation (DCAP) for maintaining consistent foreground/background appearances.
3.This paper introduce the Interaction-aware Human-Object Composition (IHOC) dataset which maybe can helpful for others research.
Weaknesses:
1.Ordinary and unremarkable writing
2.The novelty level is not too high
3.Too few bad cases

---

> ### Author Rebuttal · Authors · 2025-07-31
>
> We thank this reviewer for valuable comments. We appreciate the recognition of our IHOC dataset. We address the main concerns and questions below. (Abbreviations used: Q/W/L refer to Question/Weakness/Limitation.)
>
> ### Q1&W1: ''Ordinary and unremarkable writing''
>
> We would like to briefly summarize the logical structure of our paper:
>
> - **Background and Motivation:** We start with the practical need for compositing product images with humans in advertising scenarios (e.g., perfume ads), highlighting two essential objectives: (a) achieving natural and plausible human-object interactions, and (b) maintaining strict appearance consistency for both the person and the object.
> - **Task Definition:** We define the problem as *interaction-aware human-object composition* and explicitly state the main challenges: existing approaches often fail to deliver either natural interactions (such as human pose and gesture) or visual consistency (identity and appearance preservation).
> - **Technical Contributions:** We introduce **HOComp** to address these challenges, proposing two novel modules:
>     - **MLLMs-driven region-based pose guidance (MRPG):** A coarse-to-fine mechanism for constraining and guiding human-object interactions.
>     - **Detail-consistent appearance preservation (DCAP):** A method to preserve both the details and identities of the foreground object and the background person.
> - **Dataset Contribution:** To enable training, we introduce the **IHOC** dataset, the first large-scale dataset specifically designed for interaction-aware human-object composition.
> - **Experiments:** We conduct comprehensive experiments on HOIBench. Results demonstrate the advantages of HOComp in generating realistic interactions and visually consistent human-object compositions.
>
> We would be happy to refine our presentation if this reviewer can provide more specific information of which part (e.g., section/discussion) that we may improve on.
>
> ---
>
> ### Q2&W2: ''The novelty level is not too high''
>
> Our work has three main categories of contribution.
>
> **Idea Contribution.** In this paper, we introduce **HOComp**, the first **interaction-aware human-object composition** approach to seamlessly integrate a foreground object into a human-centric background while ensuring **harmonious interactions** and maintaining **consistent appearance** between the foreground object and the background person. Our method can be applied in areas such as automated advertising generation and digital content creation.
>
> Compared to existing methods, HOComp addresses the following shortcomings:
> (1) **Failure in natural human-object interactions:** Many image-guided composition methods fail to generate realistic human gestures/poses during human-object interaction, resulting in unnatural outcomes.
> (2) **Inconsistent appearance of humans or objects:** Existing multi-concept customization or HOI-based image generation methods regenerate both the foreground object and the background human, leading to inconsistencies in the background human's appearance.
>
> **Technical Contributions.** **HOComp** introduces two key technical contributions:
> 1. *MLLMs-driven Region-based Pose Guidance (MRPG),* which utilizes MLLMs to identify the interaction region and interaction type (e.g., holding and lifting) to provide coarse-to-fine constraints to the generated pose for the interaction while incorporating human pose landmarks to track action variations and enforcing fine-grained pose constraints.
> 2. *Detail-consistent Appearance Preservation (DCAP),* which unifies a shape-aware attention modulation mechanism, a multi-view appearance loss, and a background consistency loss to ensure consistent shapes/textures of the foreground and faithful reproduction of the background human.
>
> **Dataset/Benchmark Contributions.**
> 1. To facilitate research in interaction-aware human-object composition, we introduce the **IHOC dataset**, specifically designed for this task. It is the first dataset to contain paired pre-/post-interaction data, which is crucial for modeling realistic and coherent human-object compositions.
> 2. We also present **HOIBench**, a benchmark for comprehensive evaluation of human-object composition methods.
>
> If this reviewer can provide more specific information regarding this novelty issue, we would be happy to further discuss.
>
> ---
>
> ### W3&L1: ''Too few Bad Cases'' and More Analysis
>
> We appreciate the reviewer’s suggestion to provide more failure cases and more analysis.
>
> Our MRPG module adopts a coarse-to-fine strategy for constraining human-object interactions. At the coarse level, it leverages MLLMs to automatically identify suitable interaction types/regions via multi-stage querying. In some rare cases, MLLMs may fail to accurately predict the interaction regions. We provide a detailed analysis of these failure modes and potential solutions below.
>
> In our HOIBench experiments, MLLMs (e.g., GPT-4o) identified reasonable interaction types in **100%** of samples. The corresponding interaction region was predicted correctly in **91.33%** of cases (548/600). For the remaining **8.67%** (52/600), two main failure modes are observed:
>
> 1. **Mismatch Between Interaction Regions and Types (6% of cases):** When multiple plausible interaction types exist, MLLMs may misassign interaction regions. For example, if the object is sunglasses and the interaction type is ''hold'', the model may incorrectly assign the region around the eyes (i.e., ''wear'' action). While the generated image depicts a person wearing sunglasses with harmonious interactions and consistent appearance, the predicted interaction type is inconsistent.
>
> 2. **Incorrect Interaction Region Size (2.67% of cases):** In these cases, the predicted interaction region does not fully cover the area, including the object and the human body part, for modification. As shown in Fig.6 (main paper), this can hinder generation of correct interactions.
>
> To address these two issues, we explored the following solutions:
>
> - **Additional Input Conditions:** To alleviate the mismatching issue, we can further incorporate human pose priors (from a pose estimator) as an additional prior to MLLMs.
> For GPT-4o, the region prediction accuracy increased from **91.33%** to **96.5%**.  This improvement can be attributed to the introduction of explicit keypoint information, which provides precise localization of body parts such as the face and hands.
> - **Training with Noisy Data:** To mitigate the impact of incorrect interaction region sizes, we introduce noisy data during training.
> We first generated 1,000 accurate input data samples. In accordance with the observed error rate, we then moved the bounding boxes of 700 samples to create mismatches with the interaction types (e.g., moving the bounding box of a ''soccer ball'' with a ''kick'' action to cover the hand, which may correspond to a ''hold'' action). Additionally, we reduced the bounding boxes of 300 samples so that they no longer fully covered the interaction object or the required body movement. We fine-tuned the previously pretrained HOComp model for 3,000 steps using these noisy samples alongside the original IHOC dataset. After retraining, we re-evaluated our method using the 52 failure cases. As shown in the following tables, training with noisy data substantially improved the model's performance even when MLLMs mispredicted the interaction region.
>
> | Training Strategy      | FID↓ | CLIP-Score↑ | HOI-Score↑ | DINO-Score↑ | SSIM (BG)↑ |
> |-----------------------|-----------------|----------------------|---------------------|----------------------|---------------------|
> | Without Noisy Data    | 11.58           | 29.55                | 38.62               | 70.02                | 62.01               |
> | With Noisy Data       | **10.03**       | **29.82**            | **77.89**           | **74.29**            | **88.93**           |
>
> Qualitative results and further experimental details will be included in the supplemental materials.

---

### Comment · Area_Chair_QQbS · 2025-08-05

Dear Reviewers,

Thank you for your efforts. Please revisit the submission and check whether your earlier concerns have been adequately addressed in the author's response. If you haven't, please join the discussion with the authors actively before August 8 to let the authors know **if they have resolved your (rebuttal) questions**.

Best, AC

---

### Decision · Program_Chairs · 2025-09-17

**Decision:**

Accept (poster)

**Comment:**

The paper presents an interesting problem formulation of editing the hand object composition in an image with a generative model. Reviewers acknowledged that this paper proposes a complex yet reasonable framework along with a new dataset, and the experimental results show its effectiveness.

Initially, reviewers raise concerns about the complexity of the method, the robustness of the complex pipeline, various generalization capabilities, the evaluation of physical plausibility, and potential data bias. During the rebuttal, authors provide extensive experimental results and clarifications to address the concerns and promise to release the code. Reviewers are satisfied with this response; however, there are several remaining issues. For example, the comparison with the 3D HOI method and the lack of physically plausible measures.

Given that all technical reviewers lean toward borderline acceptance, the contribution of this paper, and the authors have demonstrated willingness and ability to address most concerns, AC recommends accepting this paper with an expectation that the issues (including ethics issues) can be fully resolved in revision.